# Single-cell analysis of human glioma and immune cells identifies S100A4 as an immunotherapy target

Nourhan Abdelfattah[1,15], Parveen Kumar[2,15], Caiyi Wang[1,3,15], Jia-Shiun Leu[1], William F. Flynn[2], Ruli Gao[4], David S. Baskin[5,6,7], Kumar Pichumani[5,6,7], Omkar B. Ijare[5,6], Stephanie L. Wood[5], Suzanne Z. Powell[6,7,8,9], David L. Haviland[10], Brittany C. Parker Kerrigan[11,12], Frederick F. Lang[11,12], Sujit S. Prabhu[11], Kristin M. Huntoon[11,12], Wen Jiang[13], Betty Y. S. Kim[11,12], Joshy George[2] & Kyuson Yun[1,14 ✉]

A major rate-limiting step in developing more effective immunotherapies for GBM is our inadequate understanding of the cellular complexity and the molecular heterogeneity of immune infiltrates in gliomas. Here, we report an integrated analysis of 201,986 human glioma, immune, and other stromal cells at the single cell level. In doing so, we discover extensive spatial and molecular heterogeneity in immune infiltrates. We identify molecular signatures for nine distinct myeloid cell subtypes, of which five are independent prognostic indicators of glioma patient survival. Furthermore, we identify *S100A4* as a regulator of immune suppressive T and myeloid cells in GBM and demonstrate that deleting *S100a4* in non-cancer cells is sufficient to reprogram the immune landscape and significantly improve survival. This study provides insights into spatial, molecular, and functional heterogeneity of glioma and glioma-associated immune cells and demonstrates the utility of this dataset for discovering therapeutic targets for this poorly immunogenic cancer.

[1] Department of Neurology, Houston Methodist Research Institute, Houston, TX, USA. [2] The Jackson Laboratory for Genomic Medicine, Farmington, CT, USA. [3] Xiangya Hospital, Central South University, Changsha, P. R. China. [4] Center for Bioinformatics and Computational Biology. Houston Methodist Research Institute Houston, Houston, TX, USA. [5] Department of Neurosurgery, Houston Methodist Neurological Institute, Houston, TX, USA. [6] Kenneth R. Peak Center for Brain and Pituitary Tumor Treatment and Research, Department of Neurosurgery, Houston Methodist Neurological Institute, Houston, TX, USA. [7] Department of Neurosurgery, Weill Cornell Medical College, New York, NY, USA. [8] Department of Pathology and Genomic Medicine, Houston Methodist Hospital, Houston, TX, USA. [9] Department of Pathology and Laboratory Medicine, Weill Cornell Medical College, New York, NY, USA. [10] Flow Cytometry Core, Houston Methodist Research Institute, Houston, TX, USA. [11] Department of Neurosurgery, The University of Texas MD Anderson Cancer Center, Houston, TX, USA. [12] The Brain Tumor Center, The University of Texas MD Anderson Cancer Center, 1515 Holcombe Blvd., Houston, TX, USA. [13] Department of Radiation Oncology, The University of Texas MD Anderson Cancer Center, Houston, TX, USA. [14] Department of Neurology, Weill Cornell Medical College, New York, NY, USA. [15] These authors contributed equally: Nourhan Abdelfattah, Parveen Kumar, Caiyi Wang. ✉email: kyun@houstonmethodist.org

Glioblastoma (GBM) is the most common and aggressive primary brain malignancy in adults[1]. The current standard of care includes maximal surgical resection followed by radiotherapy and chemotherapy with temozolomide. Unfortunately, this aggressive management is rarely curative; patients with GBM have a median survival of 15.4 months, and less than 5% of patients survive over 5 years[2]. There is clearly an urgent need to develop more effective treatments for GBM patients.

In theory, GBMs should be ideal candidates for immunotherapy, since immune cells can cross the blood-brain barrier, track infiltrating glioma cells, and selectively kill cancer cells while sparing normal brain cells. In 2019, there were over 2500 cancer immunotherapy trials involving anti-PD1/PD-L1 therapies or CAR-T cells alone[3], reflecting the promise of immunotherapy[4,5]. Unfortunately, most GBM immunotherapy trials, including vaccines, adoptive cellular therapy, CAR-T cells, and immune checkpoint blockade, have shown only modest benefits in patients with GBM[6,7]. A significant barrier to immunotherapy efficacy in GBM is the lack of tumor-infiltrating lymphocytes (TILs; <5%) but abundant immunosuppressive myeloid cells[1,8–11], making it an "immune cold" tumor. By contrast, "immune hot" tumors, characterized by abundant tumoricidal effector T cells and pro-inflammatory gene signatures necessary to mount a meaningful attack, have consistently higher response rates to immunotherapy[8,12].

Tumor-associated myeloid cells are critical regulators of tumor progression, metastasis, and immune evasion and are promising therapeutic targets[13,14]. However, elucidating their functional and molecular heterogeneity in human cancers has been challenging given a lack of lineage-specific markers and their highly plastic nature, which precludes faithful modeling and analysis in vitro[11]. Recent studies suggest that in vitro defined M1/M2 cell states do not represent in vivo tumor-associated macrophage cell states[15]. Macrophages in gliomas include embryonic yolk sac-derived brain-resident macrophages (microglia)[16] and bone marrow-derived macrophages (BMDMs) recruited to the brain after injury or tumor formation[17,18]. Glioma-associated myeloid and glioma cells secrete cytokines and metabolites that suppress TIL function. Therefore, one approach to facilitating anti-tumor immunity in GBM would be to repolarize myeloid cells to a more anti-tumorigenic state, thereby enhancing effector T cell infiltration and activation. Developing such treatments requires a comprehensive and high-resolution cellular and molecular understanding of the glioma, immune, and stromal cells that form the highly dynamic and interactive tumor ecosystem. Without gaining such insights, the application of existing immunotherapy is likely to remain ineffective in GBM patients.

In this work, we report multi-regional and -dimensional analyses of human gliomas and in doing so map GBM cellular heterotypia and spatial, molecular, and functional heterogeneity of glioma and associated stromal cells, including immune cells. We report the molecular phenotypes of glioma cells, microglia, macrophages, T cells, and pericytes within the same tumor samples in low-grade gliomas (LGGs, grades II), newly diagnosed GBMs (ndGBM, grade IV), and recurrent GBMs (rGBM, grade IV). We also demonstrate spatial heterogeneity of immune infiltrates and distinct cell:cell interaction patterns within each patient and across different patients. This integrated human glioma analysis reveals considerable spatial, molecular, and functional immune cell heterogeneity in human gliomas and nominates S100A4 as an immunotherapy target.

## Results

### A multi-regional analysis of cancer and immune cells from human glioma.
To analyze the cellular and molecular heterogeneity of human gliomas at the single-cell level in an unbiased manner, we performed single-cell RNA-sequencing (scRNA-seq) of 44 fragments of tumor tissue obtained from 18 glioma patients (2 LGG, 11 ndGBM, and 5 rGBM) (Fig. 1a, Supplementary Data 1). In ten patients, we performed a multi-regional sampling of the tumor to assess the spatial heterogeneity of cancer and immune cells in each patient tumor (Supplementary Fig. 1a, b). As shown in Supplementary Data 1, we sampled a broad spectrum of human gliomas: LGG samples included one IDH-mutant oligodendroglioma and one IDH-mutant astrocytoma, while GBMs were IDH-wildtype with mutations in common tumor suppressors and oncogenes such as TP53, PTEN, TERT, CDKN2A, CDK4, and NF1. We also performed whole exome-sequencing (WES) analysis from three GBM patients (ndGBM-01, ndGBM-02, and rGBM-01) and identified both shared and fragment-specific mutations in different regions from the same patient and among different patients (Supplementary Fig. 1c–e), consistent with previously reported inter- and intra-tumoral genomic heterogeneity of GBM[19]. For example, all three GBMs displayed loss of chromosome 10/10q and gain of chromosome 7/7q (Supplementary Fig. 1c), which are recurring copy number alterations in human GBMs[20]. Although major copy number change events were usually shared between different fragments in each tumor, there were also unique indel and mutational patterns that distinguished different fragments in each patient (Supplementary Fig. 1c–e).

### Concurrent single-cell analysis of glioma and immune cells from the same samples.
To elucidate the cellular and molecular heterogeneity of cancer and stromal cells in human gliomas, 201,986 cells from 44 samples passing all QC steps were analyzed. Unsupervised clustering using Louvain community detection revealed 12 clusters with distinct gene expression patterns (Fig. 1b–e, Supplementary Fig. 2a–c, Supplementary Data 2 and 3). Individual cells were identified as either cancer or normal based on inferred copy number alterations using the CopyKat algorithm[21] (Fig. 1b; Supplementary Fig. 3a–d), and copy number alterations were congruent with WES in three patients (Supplementary Fig. 1c). By combining CopyKat analysis and marker gene expression (Fig. 1c, d, Supplementary Fig. 2b, Supplementary Data 2 and 3), each cluster was classified as either myeloid cells (C1, C4, and C7; expressing PTPRC/CD45, ITGAM/CD11B, and CD68), glioma cells (clusters C2, C6, and C9; expressing SOX2, OLIG1, GFAP, and S100B), T cells (C3; expressing PTPRC/CD45, CD3E, CD4, and CD8A), B cells (C11; expressing CD79A and CD19), or other stromal cells (C8 pericytes expressing ACTA2 and PDGFRB; C10 endothelial cells expressing PECAM; C5 oligodendrocytes expressing OLIG2 and MBP) (Fig. 1b, c, Supplementary Data 3). As expected from previous flow cytometry, CyTOF, and single-cell analyses[15,18,22–27], the two most abundant cell types were glioma (40.5% of total) and myeloid cells (45.0% of total), while T cells constituted 9.7% of all the cells profiled (Fig. 1e, f, Supplementary Fig. 2d). Notably, more myeloid cells were present in samples from females than males (males $35.2 \pm 5.534$, females $65.627 \pm 10.7$; false-discovery rate (FDR) = 0.000041): 5 of 7 female samples contained >50% myeloid cells, while only 4 of 11 male samples contained >50% myeloid cells. In addition, 1 of 2 (50%) LGGs, 5 of 11 (45%) ndGBMs, and 1 of 5 (20%) rGBM samples contained >50% myeloid cells (Fig. 1f, Supplementary Fig. 2d, Supplementary Data 1).

### Molecular heterogeneity of glioma cells.
To elucidate the molecular heterogeneity of glioma cells, we extracted cells in clusters C2, C6, and C9 (Fig. 1b) performed de novo clustering of glioma cells and identified nine clusters (GC1-GC9) based on

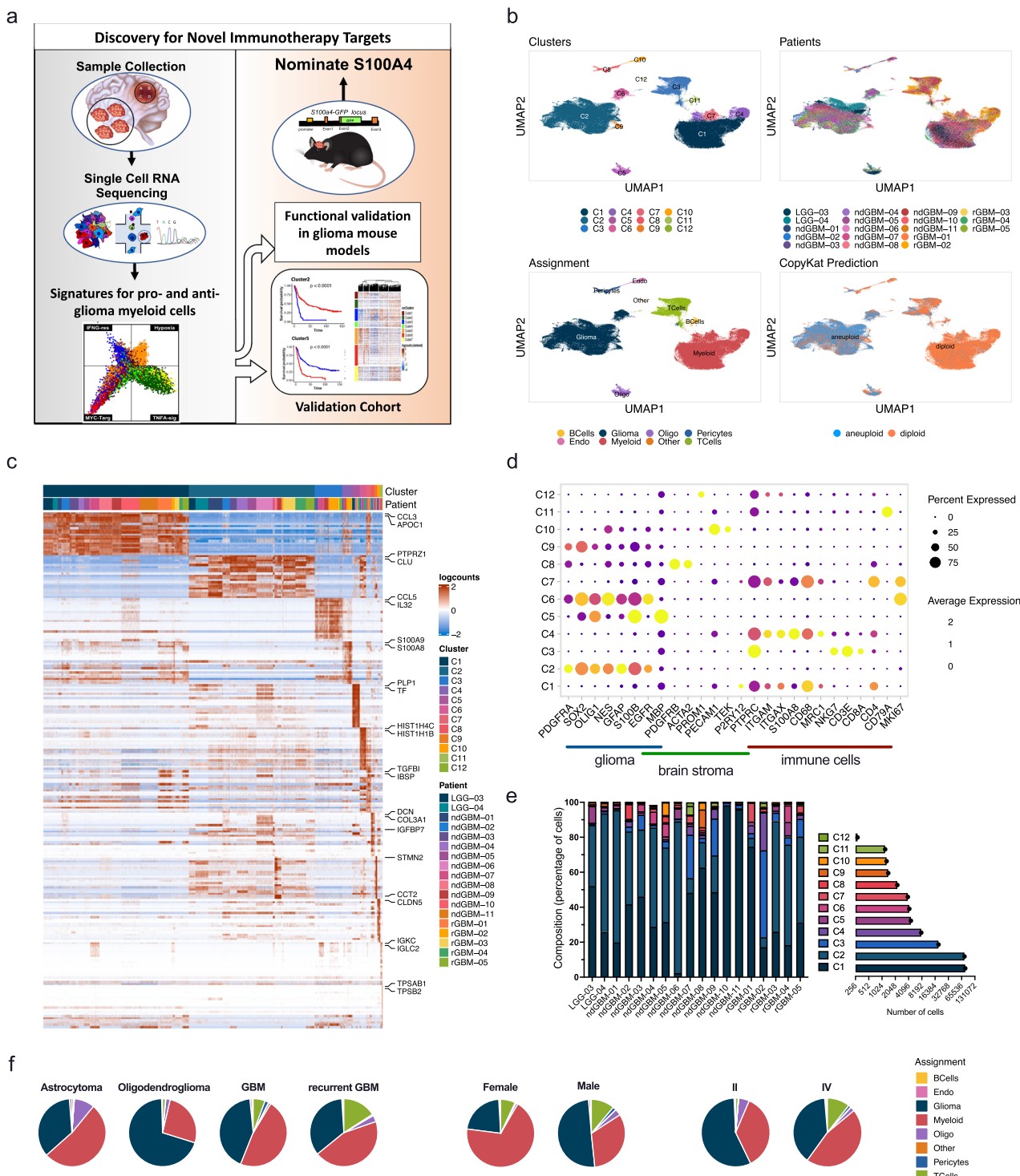

**Fig. 1 Single-cell transcriptome analysis of human glioma and immune cells. a** A schematic summary of the study design. **b** UMAP projections of 201,986 aggregate single cells from 18 patients showing the composition of different cell types in human gliomas. UMAP projections are shown by cluster numbers, by the patient, by cluster assignment, and by diploid(normal)/aneuploid(malignant) status determined by CopyKat analysis (see Supplementary Fig. 2a). **c** Top 20 differentially expressed genes in clusters, ranked by FDR, are shown in the heatmap. Gene expression values were centered, scaled, and transformed to a scale from −2 to 2. Select signature genes are highlighted on the right. **d** Dot plot showing marker gene expression for different cell types (gliomas, brain stroma (pericytes and oligodendrocytes), and immune cells). Dot sizes indicate the percentage of cells in each cluster expressing the gene, and colors indicate average expression levels. **e** Fraction of cells (y-axis) from each patient sample (x-axis) color-coded for cluster IDs as in (**b**, **c**). The numbers of cells in each cluster from all patients are also indicated in the horizontal bar graph on the right. (Also see to Supplementary Data 2). **f** Pie charts representing the percentage of cells per assignment by tumor type, sex, and tumor grade color-coded for cell type assignment. Source data for e and f are provided as a Source Data file.

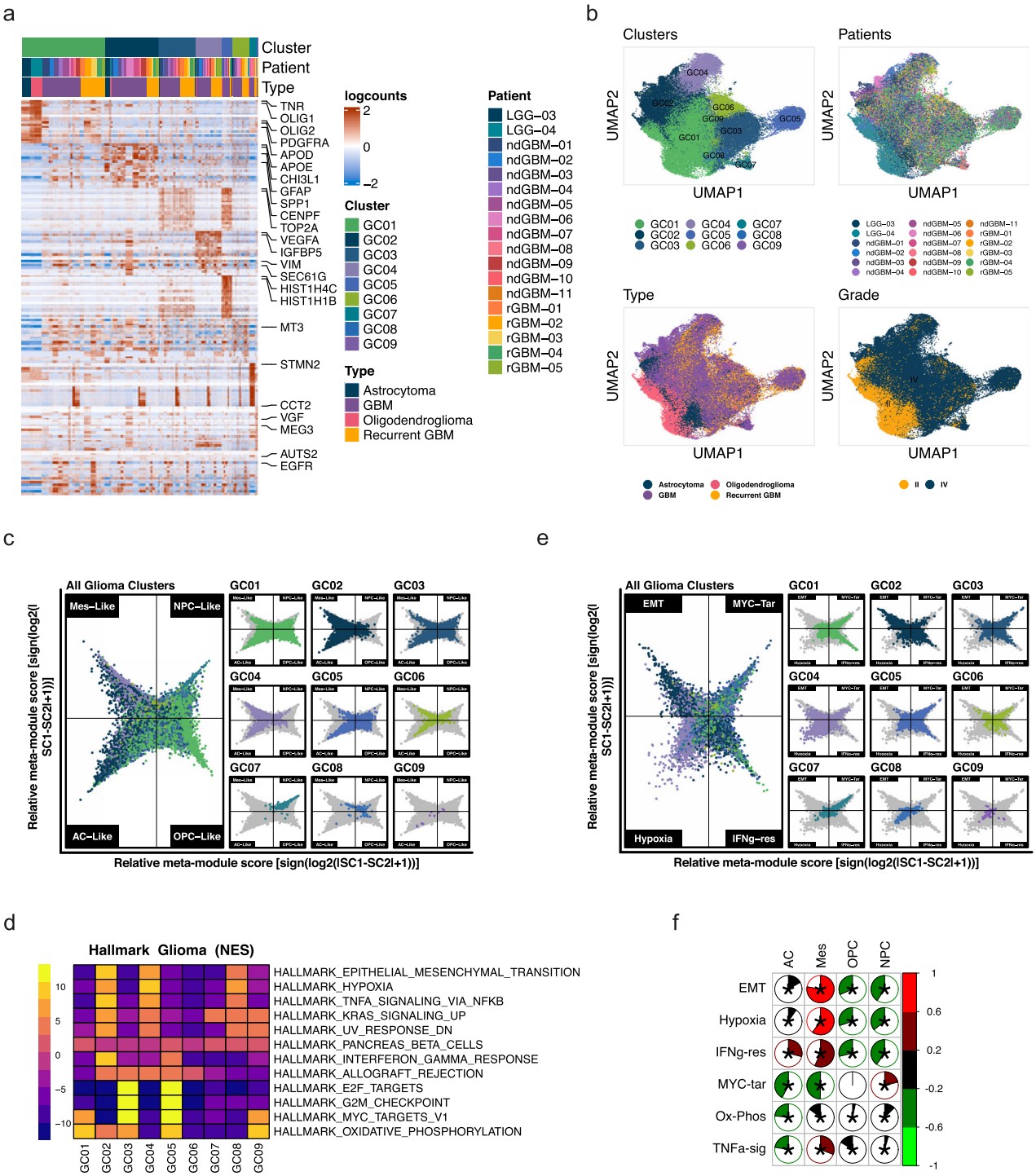

significant gene expression differences (Fig. 2a-b, Supplementary Data 4 and 5). Neftel *et al.* previously reported that GBM cells can be classified into OPC, NPC, AC, or Mes-like cell states (Neftel glioma subtypes: NG subtypes)[26]. Using their published algorithm, we projected our glioma cells onto a two-dimensional butterfly plot, with each quadrant corresponding to an NG subtype state (Fig. 2c) and there was not a discernable pattern between the glioma clusters and NG subtypes. On the other hand, each tumor contained a mixture of glioma cells in different cell states (Supplementary Fig. 3e), consistent with previous reports[26,27]. Notably, even LGGs contained heterogeneous glioma cell states: LGG-03 (a grade II astrocytoma) contained all four cell states, and LGG-04 (a grade II oligodendroglioma) contained

NPC- and OPC-like cells and a smaller proportion of AC- and Mes-like cells (Supplementary Fig. 3e).

To gain molecular insights into distinguishing features of our glioma clusters, we performed pathway analyses with cluster signature genes. Significantly enriched GSEA Hallmark pathways included epithelial–mesenchymal–transition (EMT), hypoxia, Myc-targets-v1, Interferon-gamma (IFNG)-response, TNFa-signaling-via-NFkB, and G2M checkpoint (cell cycle) hallmarks (Fig. 2d). We selected the top four pathways (EMT, hypoxia, Myc-targets-v1, and INFG-response) to generate butterfly plots by scoring each cell for its enrichment in each of the four pathways (Fig. 2e). Similar to NG classification (Fig. 2c), most glioma clusters had cells represented in multiple quadrants

**Fig. 2 Molecular characteristics of glioma cells.** Glioma cells in clusters 2, 6, and 9 from Fig. 1b were extracted and analyzed through de novo clustering. **a** A heatmap showing the top 20 differentially expressed genes in the glioma nine clusters, ranked by FDR. Gene expression values were centered, scaled, and transformed to a scale from −2 to 2. **b** UMAP projections of glioma cells only, color-coded by cluster number, patient ID, tumor type, and grade. **c** Two-dimensional butterfly plot visualization of molecular subtype signature scores per Neftel et al. Each quadrant corresponds to one subtype (mesenchymal-like (Mes-like), neural-progenitor-like (NPC-like), astrocyte-like (AC-like) and oligodendrocyte-progenitor-like (OPC-like)), and the position of each cell reflects its relative signature scores. Colors represent different clusters. **d** A heatmap representation of GSEA Hallmark Pathway gene sets showing the highest and lowest two enriched pathways in each cluster, ranked by normalized enrichment scores (NES). Adjusted p-value cutoff=0.05. Genes were pre-ranked using the Wilcoxon rank-sum test and auROC. Color bar represents NES. **e** Two-dimensional butterfly plot visualization of the top Hallmark Pathways (EPITHELIAL_MESENCHYMAL_TRANSITION, MYC_TARGETS_V1, HYPOXIA, and INTERFERON_GAMMA_RESPONSE) in different clusters, representing signature scores as relative meta-module scores. Each quadrant corresponds to one Hallmark pathway; the exact position of each cell reflects its relative signature scores in all four pathways. Colors represent different clusters shown in (**a**). Details on signature score calculation and plot generation are in the Supplementary Methods. **f** Correlogram showing Pearson correlation coefficients (r) between the top differentially enriched pathways (from **d**) and glioma molecular subtypes (Neftel et al.). Asterisks represent statistically significant comparisons (p-value < 0.05). Scale bars represent Pearson correlation (r) (red = positive correlation, green = negative correlation).

(Fig. 2e). GC2 and GC4 clusters showed significant enrichment of EMT and hypoxia. When plotted on the NG subtype butterfly plots, GC2 and GC4 cells were mostly MES-like and AC-like cells (Fig. 2c) consistent with significant correlations between EMT and hypoxia signatures to the MES-like subtype (Fig. 2f). The majority of GC7 cells fell in Myc-targets (Fig. 2e) and corresponded to NPC-like cells (Fig. 2c). GC3 and GC5 clusters showed strong proliferation hallmarks (E2F targets, G2M checkpoint, Fig. 2d) and are assigned to Myc-targets and IFNG-response quadrants (Fig. 2e) and they do not show enrichment for particular subtypes by NG subtypes (Fig. 2c). Similarly, GC1 is significantly enriched in oxidative phosphorylation (OX-PHOS) and Myc-targets (Fig. 2d) and scores high in Myc-targets and IFNG-response quadrants (Fig. 2d) but evenly spread among all four NG subtype quadrants (Fig. 2e). Together, these results suggest that molecular characteristics of glioma cell subtypes as represented by the GSEA Hallmark pathways are orthogonal to NG subtypes. Surprisingly, there was not a significant increase in MES-like glioma cells in rGBM compared to ndGBM in our cohort when only glioma cells are compared (Supplementary Fig. 3e), unlike a previous report[28].

**NK and T cell phenotypes in gliomas.** T and NK cells represented 6.4 ± 2.5% of ndGBMs, and 14.3 ± 8.9% of rGBMs, suggesting increased T cell infiltration during glioma progression (Fig. 1f, Supplementary Data 2). De novo clustering of 18,483T and NK cells (Fig. 1b, Supplementary Data 2) identified 8 clusters (Fig. 3a–c, Supplementary Data 6 and 7). Manual annotation based on marker genes revealed three CD8$^+$ T cell clusters (TC1, TC2, TC6), two CD4$^+$ T cell clusters (TC4, TC5), one naive T cell cluster (TC3), and two NK cell clusters (TC7, TC8) (Fig. 3b, c, Supplementary Data 7). CD8$^+$ T cells were most abundant in most samples (Fig. 3d), and there was no significant difference in the T cell number or subtype composition in males and females (Fig. 3e). TC4 expressed regulatory T cell (Treg) markers FOXP3, CD25/IL2RA, CTLA4, TNFRSF4/OX40, TNFRSF18/GITR, TNFRSF9/4-1BB, ICOS, and TIGIT (Fig. 3c, Supplementary Data 7) and represented 3.5 ± 3.5% of LGG T cells, 6.6 ± 1.7% of ndGBM T cells, and 8.2 ± 3.1% of rGBM T cells. Notably, PDCD1/PD1 expression was low in all samples (Fig. 3f), potentially explaining the low efficacy of anti-PD1/PDL1 inhibitors in GBM[29].

**Nine molecular subtypes of glioma-associated myeloid cells in gliomas.** Myeloid cells (including microglia and BMDM) form the largest stromal compartment in gliomas (Fig. 1). To gain molecular insights into the cellular and molecular heterogeneity of myeloid cells in gliomas, we extracted 83,479 cells in C1, C4,

and C7 and performed de novo clustering. We identified nine myeloid clusters (MC1–MC9) with unique gene expression patterns (Fig. 4a, b, Supplementary Fig. 4a, b, Supplementary Data 8 and 9). All patients contributed to each myeloid cluster (Fig. 4c, Supplementary Fig. 4b-c). As reported by others[15], glioma-associated myeloid cell subtypes in vivo did not directly correspond to in vitro-defined M0-, M1-, or M2-like macrophages using SingleR (Supplementary Fig. 4d), a reference-based cell type identification approach[30]. There was also no correlation between in vitro-defined macrophage subtypes and in vivo myeloid cells by expression correlation analysis of in vitro-defined M1, M2a, M2b, M2c, and M2d markers[31] with myeloid cluster signature genes (Supplementary Fig. 4e, f). Therefore, we manually annotated and defined distinct myeloid molecular subtypes in human gliomas using lineage markers and the molecular phenotypes of cells contained in each cluster (Fig. 4b, Supplementary Data 9).

Four clusters, MC1, MC2, MC6, and MC7, expressed previously identified microglia markers P2RY12 and TMEM119[15,32] (Fig. 4b, Supplementary Fig. 4g) and also high levels of markers BHLHE41, SORL1, SPRY1, and SRGAP28 (Fig. 4b, Supplementary Data 9). MC1 (i-Mic) expressed high levels of activated microglia markers[33] CCL3/MIP-a (macrophage inflammatory protein-1 alpha), CCL4/MIP-β, CCL3L3, CCL4L2, and CD83 (Fig. 4b, Supplementary Data 9) as well as TNF, IL1B, and NFKBIZ (Fig. 4b). In contrast, MC2 (h-Mic) expressed the highest level of CST3 (Supplementary Fig. 4g), a homeostatic microglia marker[34]. MC6 (AP-Mic) expressed both microglia and macrophage markers in addition to CX3CR1, CD86, IFNGR1, TGFB1, and B2M. MC7 (a-Mic) separated from MC1 and MC2 by differential expression levels of SPRY1, PYRY13, and microglia activation markers (Fig. 4b, Supplementary Fig. 4g).

Among the BMDM cells, MC8 (DC) represented antigen-presenting cells (APCs) expressing traditional dendritic cell markers CD1C, BATF3, and MHC-II genes (Fig. 4b, Supplementary Fig. 4h). MC4 (MDSC) expressed high levels of MIF and lower levels of mature macrophage markers CD68, CD163, CD204/MSR1, CD206/MRC1, and CD49d/ITGA4 than the remaining MCs (Fig. 4b, Supplementary Data 9). MC3 (s-Mac1) expressed high levels of monocyte marker CD14 and alternatively polarized, M2-like macrophage markers CD163 and CD204/MSR1 (Fig. 4b, Supplementary Data 9). MC5 (s-Mac2) expressed high levels of CD163, S100A4, LYZ, and markers of immune suppression: VEGFA, TGFB1, and IL10 (Fig. 4b, Supplementary Data 9). MC9 (p-Mac) expressed a high level of MKI67, indicating that BMDMs actively proliferate in situ to expand their numbers.

To determine whether these myeloid cell types are associated with particular molecular or signaling pathways, we next performed Gene Set Enrichment Analysis (GSEA; using the

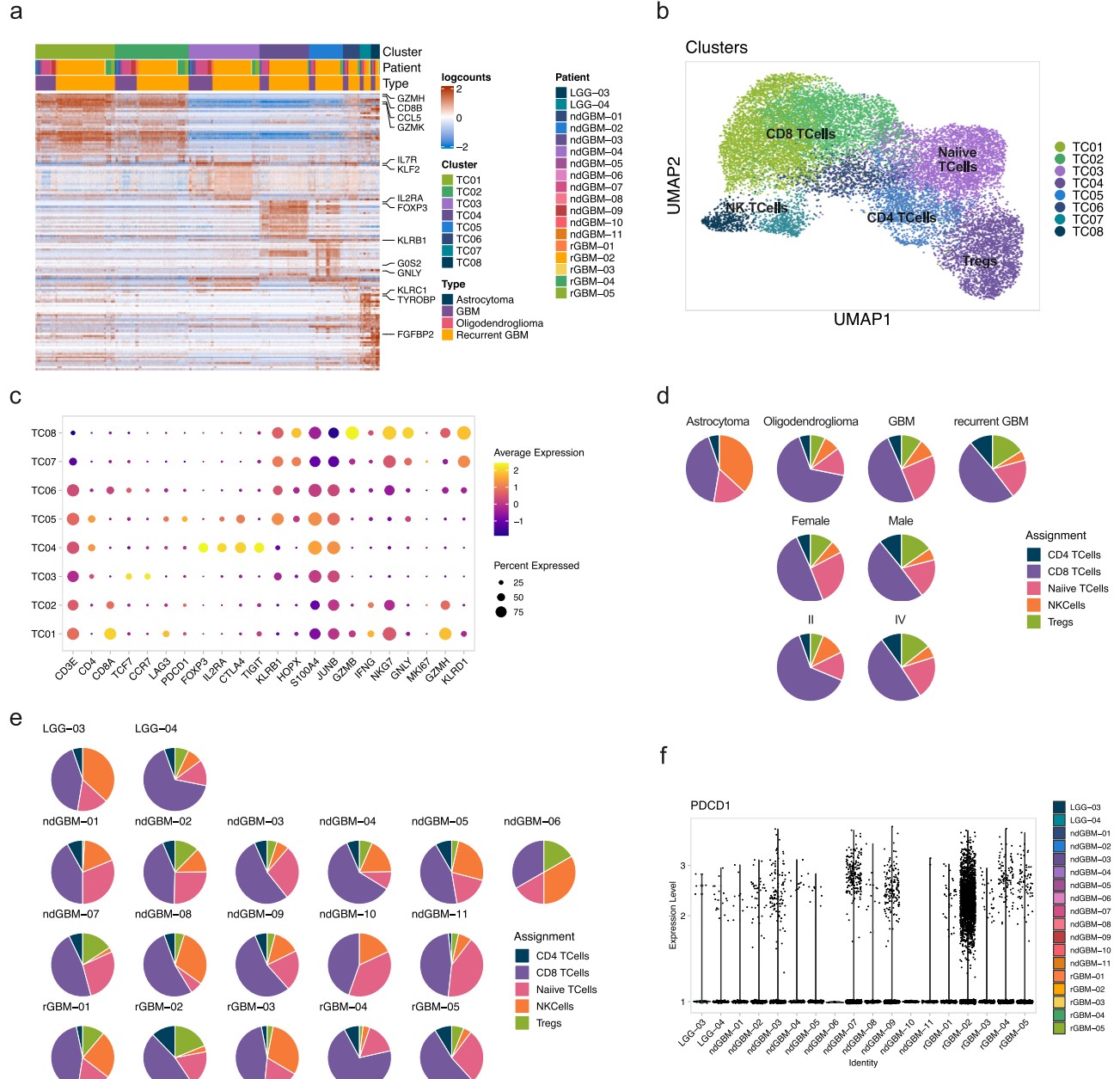

**Fig. 3 Heterogeneity of glioma-associated T and NK cells.** 18,483T and NK cells from cluster 3 in Fig. 1 were extracted and used for de novo clustering. **a** A heatmap showing the top 20 differentially expressed genes ranked by FDR in 8 clusters. Top genes are highlighted on the right. **b** UMAP projection showing de novo clustered T and NK cells. Cells are color-coded by identified clusters as in (**a**). Clusters are labeled with assigned cell types based on marker gene expression. **c** Dot plot showing the average expression of marker genes across all cells within each cluster. The size of the dot shows the percentage of cells expressing a particular gene while color shows the average gene expression levels (navy is low and yellow is high). **d** Pie charts representing the percentage of different cells types (by tumor type, patient sex, and tumor grade), color-coded for cell type assignment. **e** Pie charts representing the percentage of different cell types by the patient, color-coded for cell type assignment. **f** A violin plot showing low expression of *PDCD1* in T cells by the patient. Source data for d and e are provided as a Source Data file.

"Hallmarks" gene set from MsigDB) and Gene Ontology Enrichment Analysis with each MC signature gene (Fig. 4d, e, Supplementary Fig. 5a). Heatmap visualization revealed significant enrichment for hypoxia and EMT in MC3, MC4, and MC5 BMDMs (Fig. 4d). MDSCs (MC4) differed from macrophages (MC3, MC5, and MC9) by having lower Myc-targets-v1, OX-PHOS, IFNG-response, G2M checkpoint/E2F targets (proliferation), and adipogenesis hallmarks (Fig. 4d, e). Surprisingly, classical inflammatory hallmarks [IFN-α-response (MC3, MC6, MC8, MC9), TNFα-signaling-via-NFκB (MC1, MC3–MC5,

MC8), allograft-rejection (MC1, MC3, MC5, MC6, MC8)] were enriched in clusters of pro-tumorigenic macrophages or antigen-presenting clusters (MC3, MC4, MC5, MC6, MC8), and not MC2 and MC7 clusters (Fig. 4d–f), indicating that anti-tumorigenic macrophages respond to inflammatory signals such as IFN-γ and TNF-α in the microenvironment; however, other factors contribute to their polarization towards immune-suppressive, pro-tumorigenic phenotypes. Enrichment of OX-PHOS hallmarks in MC3, MC5, MC6, MC8, and MC9 supports current thinking that immunosuppressive macrophages utilize oxidative

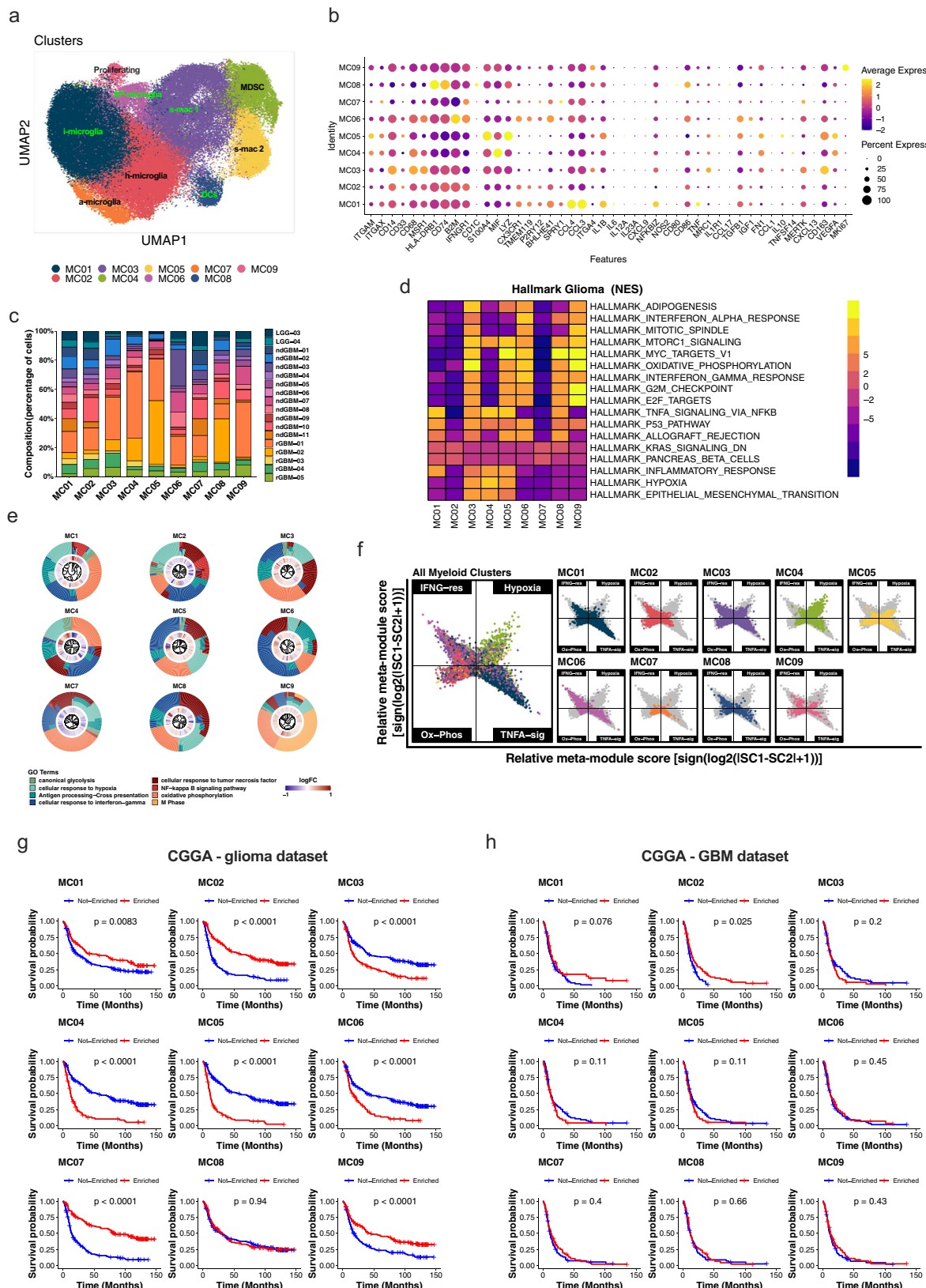

phosphorylation while inflammatory macrophages preferentially engage glycolysis[35]. However, the OX-PHOS hallmark signature in MDSCs (MC4) is equivalent to h-Mic even though it is well-established that MDSCs are immunosuppressive, suggesting that metabolic states of pro-tumorigenic myeloid cell subtypes in vivo are highly variable.

**Microglia and BMDM subtype gene signatures are prognostic.** To determine the clinical relevance and function of glioma-associated myeloid cell types, we first examined associations between each myeloid cluster signature score and patient survival in several human glioma bulk RNA-seq datasets. In the CGGA (Chinese Glioma Genome Atlas) dataset with matched RNA-seq

**Fig. 4 Nine myeloid cell subtype signatures are predictive of patient survival.** Totally, 83,479 cells from clusters 1, 4, and 7 in Fig. 1b corresponding to myeloid cells were extracted and used for de novo clustering, identifying 9 myeloid clusters (MCs). **a** A UMAP projection of de novo clustered myeloid cells. Cells are color-coded by cluster numbers. Clusters are labeled with presumed activation states: i = inflammatory, a = activated, h = homeostatic, s = suppressive, AP = antigen presenting. **b** A dot plot showing the average expression of highlighted lineage marker genes across all myeloid clusters. **c** Fraction of cells from each cluster (x-axis) color-coded by patients. **d** A heatmap showing top and bottom two enriched GSEA Hallmark Pathways in each cluster (adj. p-value cutoff = 0.05). Genes were pre-ranked using the Wilcoxon rank-sum test and auROC. **e** Gene Ontology (GO) enrichment analysis with top differentially expressed genes (DEGs) among clusters. Plots show circular dendrograms of DEGs clustered by default Euclidean distance and average linkage. The inner ring displays logFC (blue is low, red is high). The outer ring represents assigned terms. Top terms were selected based on z scores and p-values for each cluster. **f** Two-dimensional butterfly plot visualizations of top Hallmark Pathways in different clusters (TNFA SIGNALING VIA NFKB, INTERFERON GAMMA RESPONSE, HYPOXIA, and OXIDATIVE PHOSPHORYLATION), representing signature scores as relative meta-module scores. Colors represent different clusters shown in (**a**). **g**, **h** Kaplan–Meier survival curves generated with each of MC signature genes using the Chinese Glioma Genome Atlas (CGGA) dataset. **g** All glioma patients (n = 325) or **h** GBM patients only (n = 139) were stratified by positive (Enriched) or negative (Not Enriched) signature scores for each MC. Zero cell score values were used as cutoffs for positive or negative designations. P-values on graphs from univariate log-rank Mantel–Cox test (exact p-value for (**g**) (all glioma) MC2 = 1.3e−12, MC3 = 4.3e−06, MC4 = 2.04e−14, MC5 = 2e−16, MC6 = 8.62e−07, MC7 = 5.51e−15, MC9 = 3.79e−05). Also see Supplementary Data 10 for multivariate Cox regression analysis p-values for: all gliomas (MC2 = 0.04, MC3 = 0.04, MC4 = 0.0007, MC5 = 0.0003, and MC7 = 0.002) and for GBM only (MC2 = 0.02, MC3 = 0.049, and MC07 = 0.03). Source data for **c**, **g**, and **h** are provided as Source Data files.

and survival data from 325 patients (RRID:SCR_018802-mRNAseq_325 (batch 2)- http://www.cgga.org.cn/download?file=download/20200506/CGGA.mRNAseq_325.RSEM-genes.20200506.txt.zip&type=mRNAseq_325&time=20200506), gene signatures for MC3–MC6 was associated with significantly worse overall survival (Fig. 4g). In contrast, MC1, MC2, MC7, and MC9 gene signatures were associated with significantly better survival (Fig. 4g). Importantly, multivariate analysis[36] of MC signature scores, tumor subtype, gender, recurrence, *IDH* status, and *MGMT* promoter methylation status showed that MC2–MC5, and MC7 signature scores were independent prognostic indicators (multivariate Cox regression analysis p-values: MC2 = 0.04, MC3 = 0.04, MC4 = 0.0007, MC5 = 0.0003, and MC7 = 0.002; Supplementary Data 10). Notably, microglia clusters (MC2 and MC7) were associated with significantly better survival, suggesting that they are anti-tumorigenic, while macrophage/MDSC clusters (MC3–MC5) were associated with worse survival, suggesting that they are pro-tumorigenic, consistent with their marker expression patterns.

To further examine this surprising observation, we analyzed the prognostic value of MC gene signatures in GBM samples only. The MC2 gene signature was associated with significantly better survival of GBM patients in the CGGA dataset (Fig. 4h), and in the TCGA dataset (Supplementary Fig. 5b). MC3 and MC5 were associated with worse overall survival (p = 0.0055 and p = 0.016, respectively) in the TCGA dataset and the same trends were observed in the CGGA GBM dataset, although they were not significant (Fig. 4h). Multivariate analysis of the CGGA GBM only dataset with MC signature genes, gender, recurrence, *IDH* status, and *MGMT* promoter methylation status showed significant associations between MC2 and MC7 and improved overall survival and MC3 with worse overall survival (p-values: MC2 = 0.02, MC3 = 0.049, and MC07 = 0.03; multivariate Cox regression analysis; Supplementary Data 10). Together, these results indicate that the presence of specific myeloid cell subtypes is a strong independent indicator of glioma aggressiveness and patient survival.

**Validation of glioma-associated myeloid cell subtypes in an independent cohort.** To validate the reproducibility and generalizability of our categorization and signature genes, we analyzed an independent cohort of nine GBM patients from the Neftel et al. *IDH1* wildtype GBM single-cell dataset[26]. This study defined four GBM cell subtypes at the single-cell level but did not analyze the immune cell types present in their dataset. We performed de novo clustering and identified different cell types in

this dataset (Supplementary Fig. 6a). Extracting and de novo clustering 5739 myeloid cells (Supplementary Fig. 6b–d) revealed sample-specific clustering (Supplementary Fig. 6b, e), even after using Harmony[37] to remove batch effects. However, the hierarchical clustering of these cells with our MC signature genes revealed strong similarity between the identified myeloid cell subtypes (Supplementary Fig. 6f). Cells in inflammatory microglia clusters, MC1, MC2, and MC7, were segregated together and were distinguishable from cells in immunosuppressive macrophage clusters (MC3, MC4, and MC6), demonstrating the robustness and generalizability of our MC gene signatures.

**Spatial heterogeneity of immune infiltrates in human gliomas.** To examine whether immune infiltrates are spatially heterogeneous, similar to glioma cells[19], we compared the numbers and phenotypes of glioma and stromal cells from ten glioma patients from whom we collected and analyzed three to four different fragments (Fig. 5a, Supplementary Data 1 and 11). As anticipated, different fragments from the same patient contained different proportions of cancer and normal cell types, and spatial heterogeneity within a patient was observed in LGGs as well (Fig. 5a). Glioma cell subtypes (Supplementary Fig. 7a, b) and myeloid subtypes (Fig. 5c, Supplementary Fig. 7d) varied significantly from fragment to fragment. We tested whether specific clinical features associated with each sample, such as invading front or necrotic region or enhancing region correlate with any myeloid cell subtypes[15,24]. In our dataset, only three samples were annotated to originate from necrotic regions (Fig. 5b, red bars), and they did not necessarily contain more macrophages (microglia + BMDMs) than other samples. The samples from invading/infiltrating region (Fig. 5b, green bars) did contain more microglia than macrophages, supporting previous studies[15,24]. In summary, single-cell level analysis of intra-and inter-tumoral heterogeneity, using molecularly defined glioma, T cells, microglia, and BMDM cell types (Fig. 5 and Supplementary Fig. 7a, b), demonstrates significant cellular heterogeneity of cancer and immune cells in gliomas.

To determine whether regional differences in cellular composition (Fig. 5a–c, Supplementary Fig. 7) alter cell:cell interactions among different cancer and immune cells, we performed CellPhone DB analysis[38] to infer specific receptor:ligand (R–L) interactions among glioma, myeloid cells, and T cells in each fragment. As anticipated, putative R–L interactions between glioma and myeloid cells were spatially heterogeneous within each patient (Fig. 5d). Some interactions were rare or unique to a single fragment from each patient (for example, IL1R-IL1B, IL1R

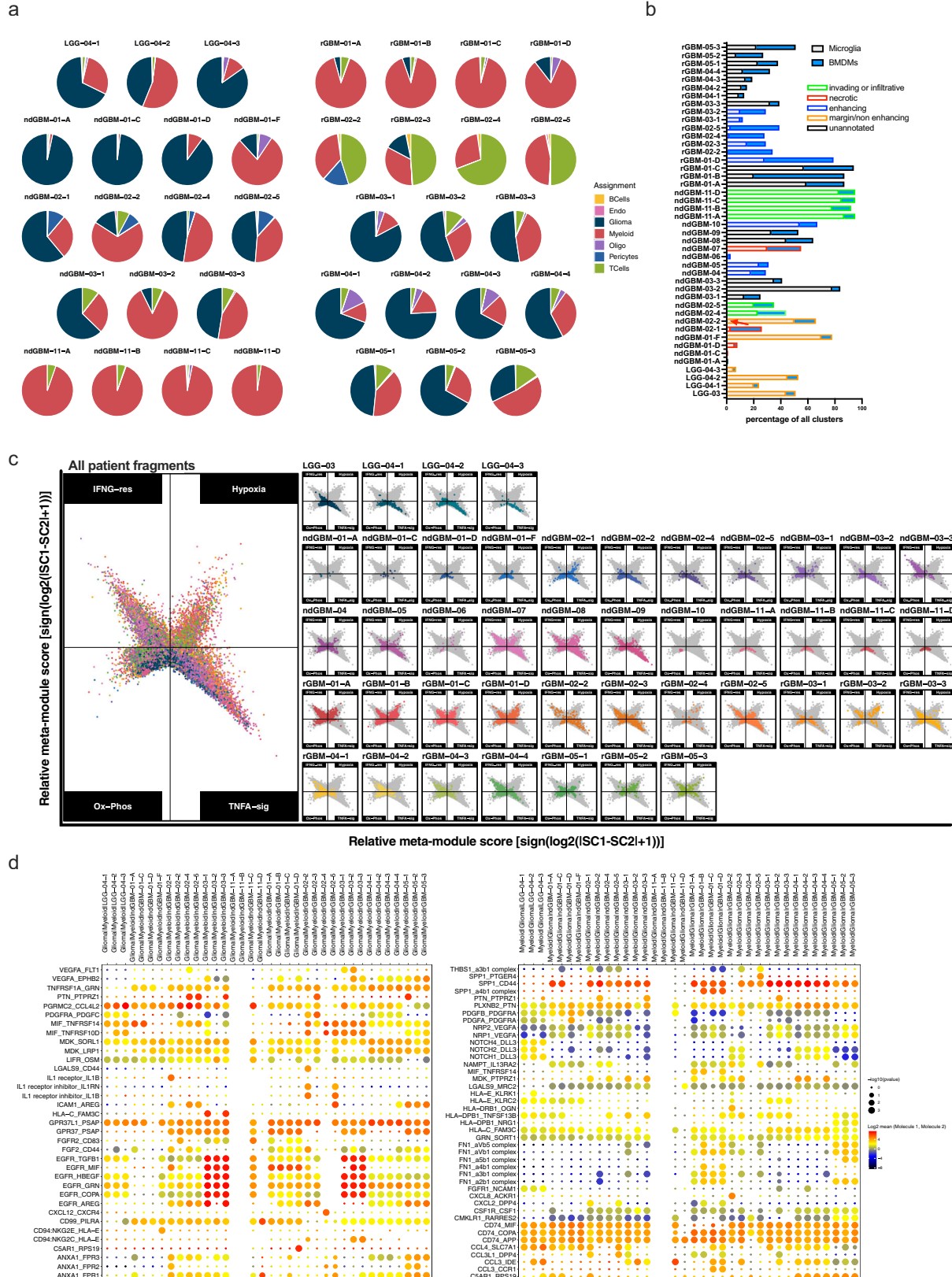

inhibitor-IL1B, HLA-C/FAM3C in glioma to myeloid signaling; CXCL8-ACKR1 in myeloid to glioma signaling, Fig. 5d). Others were robust and shared across most samples such as CD74 receptor binding to MIF, COPA, or APP as cognate ligands and SPP1-CD44 (myeloid to glioma signaling) and TNFRSF1A-GRN, PGRMC2-CCL4L2, MDK-SORL1 or LRP1, and GPR37L1-PSAP

(glioma to myeloid signaling). Interestingly, when strong EGFR signaling, with TGFB1, HBEGF, GRN, or COPA as ligands, was predicted to be present, it tended to be present in all fragments from each patient (ndGBM-03, rGBM-01, and rGBM-03). There was less R–L interaction among different cell types in LGG than in GBMs. Furthermore, bidirectional signaling between

**Fig. 5 Presence of spatially heterogeneous glioma and immune cell types results in the unique cell:cell interactions in the local microenvironment.**
Multi-regional samples from ten glioma patients were analyzed separately by fragment. **a** Pie charts representing the percentage of cells per assignment by patient fragment was color-coded for cell type assignment. **b** bar graph showing the percentage of microglia and BMDMs in different fragments. Patient fragments are highlighted with borders: red = necrotic, blue = enhancing, orange = margin or non-enhancing and green = invading or infiltrative. **c** Two-dimensional butterfly plot visualization of top Hallmark pathways in myeloid cells (TNFA SIGNALING VIA NFKB, INTERFERON GAMMA RESPONSE, HYPOXIA, and OXIDATIVE PHOSPHORYLATION) in different fragments, representing signature scores as relative meta-module scores. Colors represent different fragments. **d** Cell–cell communication analysis using CellphoneDB. Depicted are the dot plots of ligand-receptor pairs for Glioma-Myeloid (left) and Myeloid-Glioma (right) signaling across all glioma patients. Each dot size shows the −log10 p-value and color indicates the log2 mean of expression values for the listed LR pairs (y-axis) in the respective interacting cell types (x-axis, top). Dot colors represent log2 mean interaction. Only significant LR pairs, with cutoffs of p-value ≤ 0.05 and log2 mean expression value >2, are shown. The p-values were generated by CellphoneDB which uses a one-sided permutation-test to compute significant interactions. Source data for a and b are provided as a Source Data file.

glioma:myeloid and myeloid:T cell is much more abundant than signaling between T cells and glioma cells (Fig. 5d, Supplementary Fig. 7d), providing direct evidence that myeloid cells are major conduits of cell:cell interaction in the GBM microenvironment. Together, these results indicate tremendous heterogeneity of not only cancer cell phenotypes but also immune cell numbers and subtypes in different regions of gliomas. Such cellular heterogeneity results in significant differences in the local cell:cell communication among different cell types, likely amplifying local microenvironmental differences in a feedforward loop (Fig. 5d, Supplementary Fig. 7d). Whether these interactions are causes or effects (or both) of spatially heterogeneous glioma cell evolution needs to be addressed in the future.

**S100A4 is an immunotherapy target.** Having cataloged various glioma and immune cell subtypes in human gliomas, we next sought to identify immune modulatory targets. We surveyed signature genes in MC3, MC4, and MC5 pro-tumorigenic myeloid cells and TC4 (Tregs) and TC5 (exhausted T cells) to discover highly expressed genes that are shared and may be manipulated to reprogram both innate and adaptive immune cells in GBM. S100A4 stood out as a signature gene highly expressed in TC4 and TC5 and pro-tumorigenic myeloid cells (Fig. 6a, b). Immunohistochemistry on human GBM tissues showed that S100A4 was expressed in glioma infiltrating lymphocytes and macrophages (Supplementary Fig. 8), and double immuno-fluorescence analysis of human GBM and mouse glioma samples confirmed that S100A4 was co-expressed with markers of immune-suppressive macrophages (CD206 and CD163) and T cells (FOXP3, Fig. 6c). In addition, other large-scale Treg transcriptome studies have also reported S100A4 expression in Tregs[39,40]. Supportive of its role in glioma aggressiveness, elevated S100A4 expression was significantly associated with poor prognosis in glioma and GBM patients (Fig. 6d). Multivariate analysis of S100A4 expression and tumor subtype, gender, recurrence, IDH status, and MGMT status show that S100A4 is an independent prognostic factor (Multivariate Cox regression analysis p-values: all gliomas = 0.0089, GBM only = 0.0019). Therefore, we selected S100A4 as a potential therapeutic target.

To test our hypothesis, we first determined S100a4 expression in various immune cell types present in gliomas using an S100a4-GFP knock-in reporter mouse[41] (Supplementary Fig. 9a). In these mice, GFP is expressed from the endogenous S100a4 promoter instead of S100a4 when the knock-in allele is present. Flow cytometry analysis of S100a4GFP/+ (phenotypically wildtype) mouse blood confirmed S100a4/GFP expression in a subset of T cells and myeloid cells (Supplementary Fig. 9b–d). To specifically determine the functional consequence of inhibiting S100a4 in immune cells, we first deleted S100a4 from the host glioma microenvironment and orthotopically transplanted two independent syngeneic glioma tumorsphere cell lines (5459 and 2808). These primary tumorsphere lines were derived from

spontaneous S100ß-vErbB;p53 gliomas[42], and low passage cells were transplanted into age- and sex-matched C57BL6/J wildtype (B6 control) and S100a4−/− host brains (Fig. 6e). Double immunofluorescence analysis confirmed that CD45+ immune cells in S100a4−/− gliomas did not express S100A4, although S100A4 is expressed in a subset of glioma cells (Fig. 6f, Supplementary Fig. 9e). In addition, immunofluorescence analysis showed the presence of Tregs (CD3+FOXP3+) and suppressive macrophages (GFP+CD206+) in S100a4−/− host gliomas (Fig. 6f), indicating that S100a4 is not required for lineage determination or cellular differentiation of these leukocytes. An earlier study reported that S100a4−/− macrophages were compromised in their ability to migrate to sites of inflammation[43,44]; however, our flow cytometry analysis showed that macrophage numbers were equivalent between B6 and S100a4−/− host gliomas (Supplementary Fig. 9h-i). Nevertheless, S100a4−/− host mice survived significantly longer than B6 mice transplanted with the same glioma cells on the same day, in two independent glioma models (Fig. 6g). This survival benefit was associated with significantly increased CD45+ immune infiltrates, including CD4+ and CD8+ T cells, in S100a4−/− host gliomas (Fig. 6h–j, Supplementary Fig. 9e–i). Furthermore, although myeloid cell numbers were not significantly altered, T cell:myeloid cell ratios were significantly increased in S100a4−/− host gliomas due to increased T cell infiltration (Fig. 6i, Supplementary Fig. 9g), indicating effective immune reprogramming.

To functionally determine whether glioma-associated S100a4−/− immune cells are more anti-tumorigenic, we performed in vitro functional assays using T cells and myeloid cells (glioma-associated myeloid cells: GAMS) isolated from B6(S100a4+/+), S100a4+/−, and S100a4−/− host gliomas (Fig. 7a). We first measured GAM phagocytosis in vitro using pHrodo™-red labeled nanoparticles. To specifically test for cell-autonomous S100a4 function, we isolated GFP+ S100a4+/− or GFP+ S100a4−/− GAMs to only compare GAM subtypes that would normally express s100a4. S100a4−/− GAMs showed significantly more phagocytic activity than corresponding S100a4+/− GAMs (p = 0.0145, Fig. 7b). The same results were observed when wildtype GAMs (CD45+ CD11b+) were isolated from B6 host gliomas and compared to corresponding cells (CD45+ CD11b+ GAMs) from S100a4−/− host gliomas (p = 0.0004, Fig. 7c).

To functionally validate that S100a4 deletion in T cells promotes CD4 T cell activation, we measured IFN-γ secretion. FACS-sorted GFP+CD45+CD3+CD4+ TILs from S100a4+/− or S100a4−/− host gliomas were co-cultured with naïve B6 wild-type splenocytes and secreted IFN-γ in the conditioned media was measured by ELISA. S100a4−/− CD3+ CD4+ containing cultured had significantly higher IFN-γ levels compared to control cultures containing S100a4+/− T cells (S100A4+/− CD3+ CD4+) (Fig. 7d, p-value < 0.0001). Finally, we isolated GFP+S100a4+ CD4+ T cells from S100a4+/- and S100a4−/− host gliomas and co-cultured them for 4 days with in vitro activated,

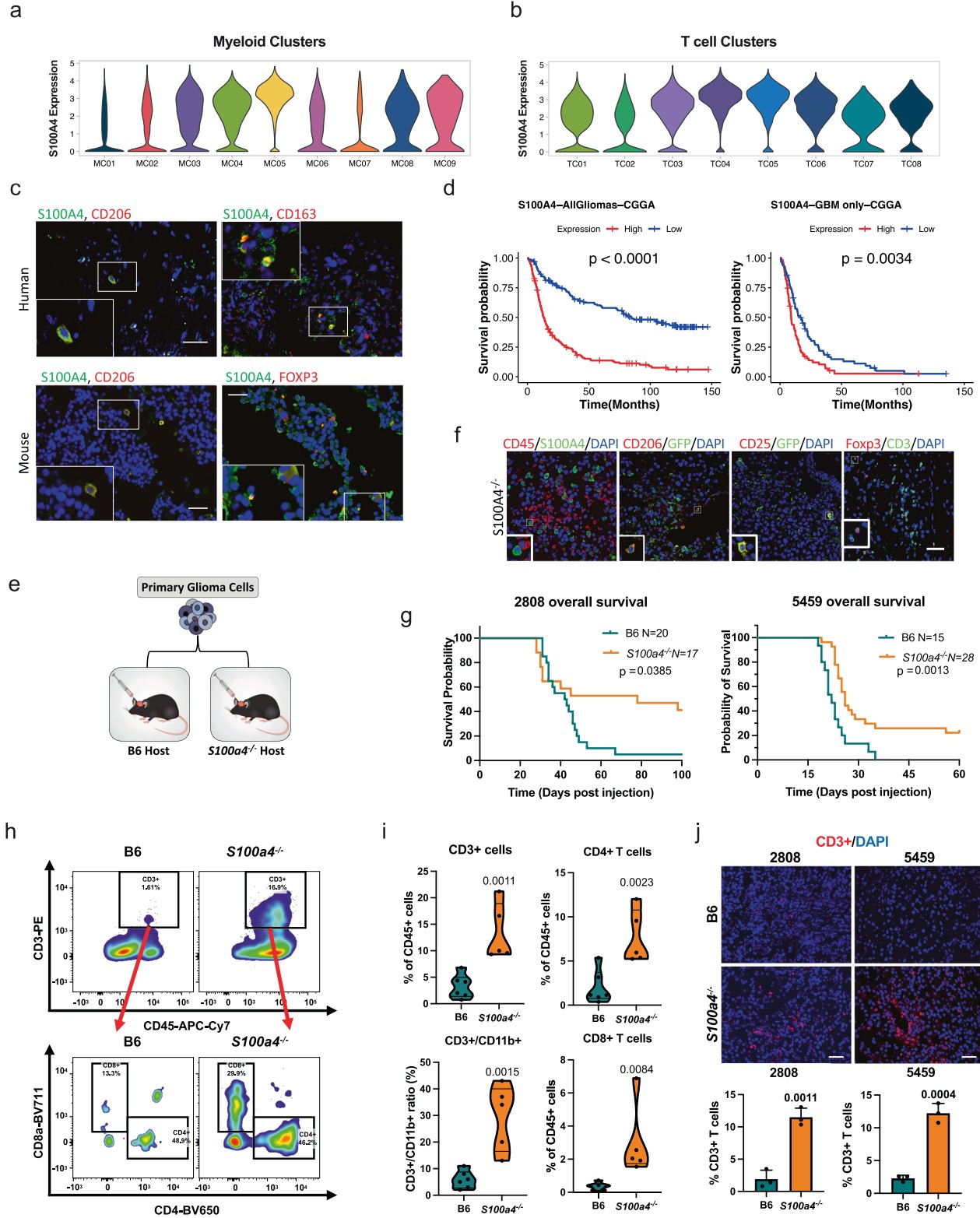

fluorescently-labeled B6 CD3$^+$ splenocytes. Flow cytometry analysis showed a significant increase in proliferation of labeled T cells in co-culture with $S100a4^{-/-}$ CD4 T cells than control CD4 T cells ($p = 0.0024$, Fig. 7e). Taken together, our functional analyses of GAMs and TILs from $S100a4^{-/-}$ host gliomas provide compelling evidence that $S100a4$ functions in glioma-associated immune cells in a cell-autonomous manner to suppress the immune response and promote glioma growth.

## Discussion

It has been widely accepted that molecular and cellular hetero-geneity of glioma cells, both between patients and within the same patient, poses a major treatment challenge and underpins the failure of targeted therapy and chemotherapy, resulting in invariable GBM recurrence[19,26,27]. However, the extent to which spatial and cellular heterogeneity of stromal cells, particularly the myeloid cells that can constitute more than half of the tumor cells

**Fig. 6 S100A4 promotes immune suppression and glioma growth. a**, **b** Violin plots showing *S100A4* expression levels in human glioma-associated myeloid cells (**a**) with high expression in immune-suppressive MCs (MC3–MC5) and glioma-associated TCs (**b**) with highest expression in TC04 and TC05, Treg and exhausted CD4 T cells. Violin plots are color-coded by corresponding myeloid and T cell clusters in Figs. 3 and 4). **c** Representative double immunofluorescence images showing co-expression of CD206, CD163, or FOXP3 (red) with S100A4 (green) in human GBMs (top) and mouse glioma (bottom). *n* = 6 patients, *n* = 3 B6 mice. Scale bar: 50 μm. **d** Kaplan–Meyer survival curve with differential S100A4 expression levels in all glioma patients (left: *n* = 325, 163 high and 162 low) and GBM patients only (right: *n* = 139, 67 high and 72 low) from the CGGA dataset, stratified by median *S100A4* expression level. *P*-values from Log-rank Mantel–Cox test. **e** Functional validation experimental design: mouse primary glioma tumorspheres isolated from spontaneous *S100ß-vErbB;p53* glioma models (5459 or 2808) were intracranially injected into sex- and age-matched B6 or *S100a4⁻/⁻* host mice. **f** Representative doubles IF images showing co-expression of CD45, CD25, or CD206 with S100A4/GFP in mouse tumors from *S100a4⁻/⁻* hosts. *n* = 3 each. Scale bar: 50 μm. **g** Kaplan–Meier survival curves showing significant survival extension of *S100a4⁻/⁻* host mice, compared to B6 hosts, intracranially injected with the same primary glioma tumorsphere cells: 5459 (B6 *n* = 15, *S100a4⁻/⁻* *n* = 28) or 2808 (B6 *n* = 20, *S100a4⁻/⁻* *n* = 17). *P*-values from Log-rank Mantel–Cox test. **h** Representative dot plots from flow cytometry analysis of tumor-infiltrating T-cells in B6 vs. *S100a4⁻/⁻* host mice. **i** Flow cytometry analysis *n* = 6 (B6) and *n* = 5 (*S100a4⁻/⁻*) mice. All pairwise analyses were performed using two-tailed *t*-tests. **j** Representative Immunofluorescence images of CD3+ T cells in B6 control and *S100a4⁻/⁻* host gliomas. CD3+ (red) and DAPI (all nuclei in blue) were counted from three fields/sample and three samples/type. Error bars represent SD. *P*-values represent two-tailed *t*-tests. Scale bar: 100 μm. Source data for **d**, **g**, **i**, and **j** are provided as a Source Data file.

(Fig. 1b, e, f), contributes to disease progression and aggressiveness is less well understood[15]. Here, we isolated and analyzed over 200,000 single cells from 44 samples from 18 low- and high-grade glioma patients. Through multi-region sampling and single-cell RNA-sequencing analysis, we demonstrate extensive heterogeneity in the numbers and types of immune cells present in different regions of the same patient tumor and among different patients. Importantly, we show that five specific myeloid cell subtype gene signatures (MC2–MC5, and MC7) are independent prognostic indicators of glioma patient survival, independent of known covariates of glioma patient survival, such as IDH mutation and MGMT methylation status. This observation clearly indicates the clinical relevance of specific myeloid cell subtypes and underscores the need for more precise and context-sensitive intervention strategies that specifically target harmful (tumor-promoting) myeloid cells while sparing helpful (inflammatory/tumor-suppressive) cells.

By performing putative receptor-ligand binding analysis in each sample, we provide evidence that the presence of phenotypically distinct myeloid cells in the local microenvironment can result in significant differences in cell:cell communication. In addition, we identify specific receptor-ligand pairs that mediate cell-type-specific communications in human gliomas. For example, PTPRZ1/PTN represents one of many spatially heterogeneous glioma:myeloid signaling nodes (Fig. 5d), and PTPRZ1/PTN signaling has been reported to promote glioma stem cell maintenance and glioma growth[45]. SPP1(osteopontin)/CD44 signaling is observed in most GBM samples, and SPP1 is a well-characterized promoter of glioma aggressiveness. Pietras et al. showed that Osteopontin-CD44 signaling promotes GBM stem cell maintenance in the perivascular region[46], and Wei et al reported that glioma cells secrete osteopontin to promote macrophage infiltration and M2-like polarization[47], contributing to immune suppression and glioma growth. On the other hand, Szulzewsky et al. reported that gliomas implanted in *SPP1*−/− mice grow more aggressively and result in shorter survival[48], suggesting a complex and context-dependent function of *SPP1*. We detected potential SPP1-CD44 signaling among T cells/glioma and T cell/myeloid as well as myeloid/glioma cells (Fig. 5d and Supplementary Fig. 7d), and it will be important to elucidate the functional consequences of inhibiting the SPP1-CD44 (and other SPP1 involving signaling) axis in each cell type to fully understand its various functions in different cell types. Finally, we provide evidence that reciprocal signaling between myeloid:glioma and myeloid:T cells is much more robust, both quantitatively and qualitatively than glioma:T cell signaling (Fig, 5d, Supplementary Fig. 7d), indicating myeloid cells as major

conduits for cell:cell signaling in glioma microenvironment. While this manuscript was in revision, Hara et al., reported that macrophage-derived OSM induces mesenchymal transition in glioma cells[49].

There is a growing consensus that targeting immunosuppressive macrophages may be critical to improving immunotherapy efficacy[13,14]. For example, Goswami et al., proposed targeting CD73+ macrophages in combination with anti-PD1 and anti-CTLA4[25] may be effective in treating GBM. Unfortunately, we did not detect significant CD73/NT5E expression in any myeloid cells in our dataset nor those of Neftel et al. Independently, we identified five (when confounding factors such as IDH mutation and MGMT methylation status are accounted for) macrophage subtypes that are prognostic of glioma patient survival. Both activated (MC7) or homeostatic (MC2) microglia were associated with improved overall survival, while MDSC (MC4) and suppressive BMDM signatures (MC3, MC5) were associated with worse survival. Interestingly, a microglia subtype (MC6) is associated with worse survival (Fig. 4h), although it did not reach significance when other variables were accounted for. However, it is clearly not associated with better survival, unlike other microglia subtypes., indicating that not all microglia subtypes are inflammatory or anti-tumorigenic. In other words, cell of origin does not correlate absolutely with the myeloid subtype function. Similarly, the proliferating macrophage cluster (MC9) was associated with better survival (Fig. 4h), but not significantly when other confounding factors were accounted for. These results are somewhat inconsistent with an earlier report by Muller *et al.*, who reported that macrophage ontogeny is the major driver of their functional phenotypes[15]. This difference most likely stems from the large sample size in our study: we analyzed 83,479 myeloid cells (~15× more), enabling us to resolve macrophage and microglia subtypes at a higher resolution. The robustness and generalizability of our gene signatures were validated in an independent, published, single-cell dataset of nine GBM patients (Supplementary Fig. 6). Furthermore, three subtype gene signatures (MC2, MC3, and MC7) were confirmed to independently stratify patient survival even among grade IV GBM patients.

In addition, we identified molecular signatures that distinguish MDSCs (MC4) from immune-suppressive macrophages (MC3 and MC5) at the single-cell RNA level, providing molecular insights into these related but distinct cell types. For example, while immune-suppressive macrophages showed a strong OX-PHOS signature, MDSCs did not. The current understanding in the field is that inflammatory macrophages preferentially engage in glycolysis, while immunosuppressive macrophages utilize OX-PHOS[35]. Therefore, targeting OX-PHOS is considered a viable

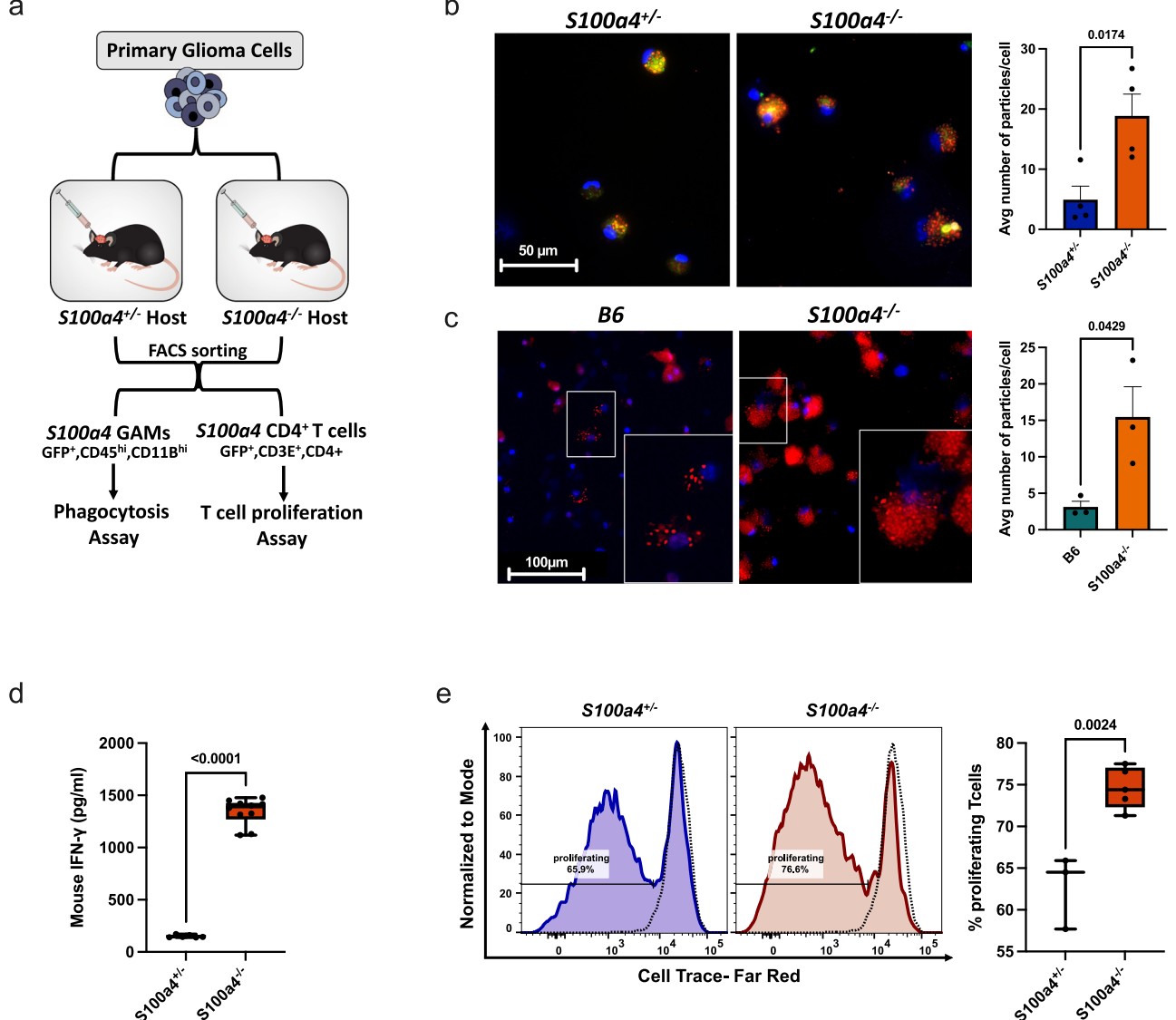

**Fig. 7 *S100a4* deletion enhances phagocytosis in myeloid cells and increases T cell activation. a** A schematic summary of experimental design. *S100a4* expressing GAMs (CD45hiCD11bhi) and CD4+ TILs (CD45+CD3+CD4+) were FACS sorted by GFP expression, and used in in vitro functional assays. **b** Fluorescence images showing phagocytosis of pHrodo™ labeled (red) nanoparticles in FACS-sorted GFP+CD45hiCD11bhi GAMs from *S100a4* heterozygous or homozygous hosts. An average number of particles/cell were calculated (*n* = 4 tumors each). *P*-value from two-tailed student *t* test. Error bars represent SEM. Scale bar: 50 μm. **c** Fluorescence images showing phagocytosis of pHrodo™ labeled (red) nanoparticles in FACS sorted CD45hi CD11bhi tumor-infiltrating GAMs from B6 or *S100a4*−/− hosts. An average number of particles/cell were calculated (*n* = 3 tumors each). *P*-value from two-tailed student *t* test. Error bars represent SEM. Scale bar: 100 μm. **d** Box and whiskers plot showing IFN-γ levels measured by ELISA in FACS-sorted GFP+ CD45+CD3+CD4+ tumor-infiltrating T cells from *S100a4* heterozygous or homozygous hosts. *P*-value represents two-tailed student's *t* test (exact *p*-value = 1.8124E−12). Whiskers represent minimum and maximum values, the line inside the box represents the mean and the box extends from the 25th to 75th percentiles. *n* = 3 (heterozygous) and *n* = 5 (homozygous) gliomas, two experimental replicates from each tumor. **e** T cell proliferation assay using dye dilution. T cells were isolated from B6 spleens, labeled, and stimulated with CD3/Cd28 dynabeads then cocultured with FACS sorted GFP +CD45+CD3+CD4+ tumor-infiltrating T cells for 4-days. Dotted lines represent unstimulated T cells. *P*-value from two-tailed student *t* test. Whiskers represent minimum and maximum values, the line inside the box represents the mean and the box extends from the 25th to 75th percentiles. *n* = 3 (heterozygous) and *n* = 5 (homozygous) gliomas. Source data for **b**–**e** are provided as a Source Data file.

strategy to reprogram immune-suppressive myeloid cells; however, our results suggest that the therapeutic window for OX-PHOS inhibitors may not exist for GBM since MDSCs may not be vulnerable to such inhibition. Our study also demonstrated that microglia and BMDMs exist in multiple cellular states and the gene signatures that distinguish them molecularly not only include previously identified environmental factors such as hypoxia but also inflammatory cytokines, such as IFNG and TNFa (Fig. 4e). Activation and secretion of IFNG and TNFa are

strongly associated with inflammation but our data suggest that these same signals may promote monocyte maturation into immune suppressive macrophages since MC3–MC5 show the strongest Hallmark signatures of TNFa signaling and IFNG-response gene sets.

We propose that immune-suppressive macrophage subtype signatures represent a rich source of therapeutic targets to enhance the efficacy of existing immunotherapies and to inform the next generation of immunotherapies to reprogram the

myeloid compartment. As a proof of principle, we selected and functionally tested *S100A4* as an immunotherapy target, based on its particularly high expression in immune-suppressive macrophages and T cells. S100A4 is a small calcium-binding protein that can function both extra- and intra-cellularly to affect multiple biological processes depending on its binding partners[50–52]. It is also considered an alarmin or damage-associated molecular pattern molecule upregulated by damaged or stressed cells[53]. As a secreted protein, it can bind to RAGE, TLR4, or EGFR family members[54], and can potentially affect immune signaling. As an intracellular signaling protein in the cytoplasm and nucleus, it can regulate diverse processes depending on binding partners and cellular contexts, such as invasion[55,56], stemness[42,57], angiogenesis[58,59], and p53 function[60]. *S100a4* is not an essential gene, since *S100a4*$^{-/-}$ mice are viable and fertile[41–43], supporting its favorable safety profile as a potential anti-cancer therapeutic. Although an earlier study reported increased tumor formation in *S100a4*$^{-/-}$ mice[61], we did not observe spontaneous tumors in our *S100a4*$^{-/-}$ colony in over a decade. In the immune system, *S100a4* is not required for T cell inflammatory responses[62], but *S100a4*$^{-/-}$ macrophages have compromised chemotaxis in vitro and infiltration to inflamed sites in vivo[43,44]. S100A4 is also implicated in several chronic inflammatory diseases including rheumatoid arthritis, asthma, and allergies[50]. High *S100A4* expression is associated with poor survival in glioma (Fig. 6g) as well as in breast[63,64], bladder[65], head and neck, and pancreatic[66] cancers, and S100A4 is necessary for breast cancer metastasis[67]. More recently, S100A4 has been shown to protect MDSCs from apoptosis[68]. We reported previously that S100A4 is necessary for glioma stem cell self-renewal and proneural–mesenchymal transition[42]. Together, previous studies indicate that S100A4 function is highly context-dependent, and in cancer, it is associated with tumor aggressiveness.

The results presented here establish a critical role for *S100a4* expression in GBM associated T cells and macrophages in promoting immunosuppression and glioma growth. *S100a4*$^{-/-}$ CD11b+ GAMs have increased phagocytic activity, which is critical to generate anti-tumor immunity. In addition, *S100a4*$^{-/-}$ CD4+ T cells stimulate T cell proliferation and secrete high levels of IFNG compared to S100a4+/− CD4+ T cells. As a consequence of reprogrammed myeloid and T cells, *S100a4*$^{-/-}$ glioma-bearing mice live significantly longer than B6 wild-type host mice, validating the potential of S100A4 as an immunotherapy target in GBM.

In summary, we anticipate that this large human glioma single-cell RNA-seq dataset will be a useful resource for the wider glioma and tumor immunology community. It can be mined to identify therapeutic targets and better understand the molecular and functional heterogeneity of glioma and immune cells and compared to other cancer types. Furthermore, it can be used to prioritize molecular targets that are commonly activated across most samples, anticipate on- and off-target effects based on cell-type-specific expression patterns, and analyze specific cell:cell interactions and expression patterns to design effective combination therapies.

## Methods

**Human tumor specimen collection**. Human tumor tissue was obtained under Institutional Review Board (IRB)-approved protocols (Pro00014547) at Houston Methodist Hospital, Houston, Texas and MD Anderson Cancer Center (PA 19-0661) in accordance with national guidelines. All patients signed informed consent during clinical visits before surgery and sample collection. Patients did not receive compensation in return for their participation in this study. The clinical characteristics of the patient samples are described in Supplementary Data 1.

**Whole exome sequencing analysis**. DNA was extracted from frozen tumor tissues using the DNeasy® Blood & Tissue (#69504; Qiagen, Hilden, Germany)

standard protocol. Whole exomes were captured using the Agilent V6 exome kit at BGI and sequenced on the DNBseq (100bpPE reads) platform. Exome sequencing reads were mapped to human reference genome GRCh38 using BWA V0.7.1 (RRID:SCR_010910), and duplicates were removed using Picard V1.95 (RRID:SCR_006525). The resulting BAM files were realigned around indels and recalibrated for base quality using GATK V3.5-0 (RRID:SCR_001876) with known variant sites from dbSNP-144 and the 1000 Genomes project (RRID:SCR_008801). Somatic mutations were called as tumor-normal pairs using MuTect2 (GATK V3.5-0) (RRID:SCR_000559). The SnpEFF package V4.3 (RRID:SCR_005191) was used to annotate the somatic mutations, and only variants annotated as high or moderate impact were used for downstream analysis. The Sequenza algorithm V2.1 (RRID:SCR_016662) with default parameters was used to determine the copy number profiles of bulk exome datasets. ggplot2 V3.3.3 (RRID:SCR_014601) and ComplexHeatmap V2.7.8.100 (RRID:SCR_017270) packages were used for visualization.

**Single-cell RNA-seq data collection**. Tumor tissues were transported on ice in DMEM/F-12 medium immediately following surgical resection. Tumors were rinsed in PBS to remove circulating immune cells in the blood (#21-40-CMR; Corning Inc., Corning, NY) and cut into small pieces and for histological analysis. Using a scalpel, the remaining tumor was minced into smaller fragments and digested in an enzyme cocktail (10% collagenase (#17104019; Gibco, Thermo Fisher Scientific, Waltham, MA): Accutase) and incubated for 10–30 min at 37 °C. Digestion cocktails were removed and replaced with DMEM/F-12:1:1 medium and gently titrated to make single-cell suspensions. Single cells were filtered using a 40 μM nylon mesh (#352340; Falcon, Thermo Fisher Scientific) to remove residual clumps. Dead cells were removed using either fluorescence-activated cell sorting (FACS) or using a dead cell removal kit (#130-090-101; Miltenyi Biotec, Bergisch Gladbach, Germany). For FACS, cells were sorted using the FACS Aria II sorter (BD Biosciences, Franklin Lakes, NJ). Dead cells were stained with DAPI or Fixable Viability Dye eFluor™ 450(FVD) (#65086318 l; Invitrogen, Carlsbad, CA). Doublets were excluded based on forward and side scatter and live cells were obtained by gating on viable cells. Sorted live cells were collected in 4 °C prechilled tubes containing 100% FBS and immediately spun down for cell counting and loading onto the 10× Chromium controller, targeting 6000 cells for capture per well.

Single-cell RNA sequencing libraries were generated using the Chromium Single Cell 3′ Library & Gel Bead Kit V1 or V3 and Chromium Single Cell 3′ Chips according to the manufacturer's instructions. In brief, all single-cell samples and required reagents were loaded on a 10× Chromium controller for droplet generation, followed by reverse transcription in the droplets, cDNA amplification, fragmentation, adapter, and index addition following the manufacturer's instructions. Barcoded single-cell transcriptome libraries were sequenced with 100 bp paired-end reads on HiSeq 4000 or BGI DNBseq platforms.

**Single-cell RNA sequencing analysis**. Raw Illumina sequencing reads were aligned to GRCh38 (human) using Cell Ranger V5 software (RRID:SCR_017344) with default parameters. Subsequently, genes were quantified as UMI counts using Cell Ranger and initially visualized using Loupe Browser V5 (RRID:SCR_018555). Downstream analysis was performed on filtered feature counts generated by Cell Ranger, and low-quality single cells containing <500 expressed genes or >20% mitochondrial transcripts or >50% ribosomal transcripts were removed. Additionally, genes expressed in fewer than three single cells were removed. We identified potential single-cell doublets using DoubletFinder V2.0.3 (RRID:SCR_018771), with an expectation of a 7.5% doublet rate assuming Poisson statistics, as per the developer's code on GitHub. Following the removal of low-quality and doublet cells, single cells were normalized and clustered using Seurat V4.0.0 (RRID:SCR_016341) and batch-corrected using Harmony V1.0. Single-cell gene expression counts were normalized to the library size and log2-transformed. We applied principal component analyses to reduce the dimensionality of the data using the top 2000 most variable genes in the dataset. Computed principal components were batch corrected for variations between patients and sex using the Harmony R package V1.0. We used batch-corrected PCs as input for Louvain-based graphing and chose resolution parameters between 0.1 and 1 depending on the single-cell datasets. Seurat V4.0.0 (RRID:SCR_016341) was used to identify cluster-specific marker genes and visualization with dot and feature plots. The genes specifically expressed in each cluster were examined to identify the cell types. Separately, we also used the reference-based R package SingleR V1.8[30] to identify the sub-cell types in an unbiased marker-free manner for T cells. SingleR compares expression profiles of single cells against reference transcriptomes of pure cell types to infer the cell of origin.

**Classification of the tumor and normal cells (CopyKat)**. All cells were classified as either normal or tumor based on the genome-wide copy number profiles computed from the gene expression UMI matrix using the Bayesian segmentation approach, CopyKat[69] V0.1.0. Aneuploid single cells with genome-wide copy number aberrations were predicted to be tumor cells. Diploid cells were predicted to be normal stromal or immune cells. The CopyKat-based predictions were further confirmed by single-cell gene expression profiles, where known GBM tumor markers including *SOX2* and *OLIG2* are highly expressed in predicted tumor cells,

and known immune cell markers including *PTPRC*, *CD3D*, and *CD68* are highly expressed in predicted immune cells.

**Pathway enrichment analyses.** We used different approaches to identify and visualize enriched pathways in our subsets.

(1) *Gene ontology enrichment analysis (GO).* To identify enriched molecular pathways based on differentially expressed genes (DE genes), over-representation analysis was performed on DE genes from each cluster using g:Profiler V0.2.0 (RRID:SCR_006809). Genesets from Gene Ontology (GO) biological processes, Reactome, and Kyoto Encyclopedia of Genes and Genomes (KEGG) (RRID:SCR_012773) were used. GOplot V1.0.2 was used to visualize the results.

(2) *Gene set enrichment analysis (GSEA).* We used fGSEA V1.14.0 (RRID:SCR_020938) R package to test for enrichment of the Hallmark genesets downloaded from MsigDB (RRID:SCR_016863, msigdbr R package V7.2.1). For input, we used either z-score statistics from Seurat DE analysis or pre-ranked gene lists generated using a fast Wilcoxon rank-sum test (presto R package V1.0.0 "github.com/immunogenomics/presto").

(3) Select gene set signature scoring. To generate the butterfly plots in Fig. 4f, we first selected the four most significantly enriched pathways generated from GSEA analysis across our clusters. Then, we adopted the method developed by Neftel et al.[26] to obtain single-cell scores using the "score" function from JLaffy/scrabble R package[26]. For each gene set, a signature score (SC(i)) was calculated for each cell (i) by first quantifying the averaged relative expression of the genes in said geneset (Er) followed by normalization by subtracting the averaged relative expression of a control gene set (Ercontrol): $SC(i) = Er(i) - Ercontrol(i)$. The control gene set was defined as described in Neftel et al.[26]. The exact position of each dot on the butterfly plot was calculated using scrabble::hierarchy() function in R using $[sign(SC1-SC2)*log2(|SC1-SC2|+1)]$.

**Comparison to in vitro-defined macrophage subtypes.** To determine whether glioma-associated myeloid cells could be classified into in vitro-defined macrophage subtypes, we designed meta-modules based on known genes upregulated in M1, M2a, M2b, M2c, and M2d macrophages[31] (Supplementary Fig. 4f). Signature scores for each meta-module were calculated using the JLaffy/scrabble R package as above, and the results were visualized using boxplots (ggplot2 R package) (Supplementary Fig. 4e).

**Assignment of GBM subtypes.** Meta-modules defined by Neftel et al.[26] was used to assign glioma molecular subtypes (MES1-like, MES2-like, NPC1-like, NPC2-like, AC-like, and OPC-like) to our human. For our analyses, we collapsed the MES1 and MES2 groups into one group of MES-like cells, and similarly the NPC1 and NPC2 into one group of NPC-like cells. We used the scrabble package to calculate meta-module scores using the "score" function and develop two-dimensional plots representing cellular states, where each quadrant corresponds to one cellular state. The exact position of each dot was calculated using scrabble::hierarchy() function in R. Results were visualized using ggplot2.

**Myeloid cluster signature gene validation in an independent cohort of GBMs.** For the validation glioma dataset, we obtained scRNA-seq data from the Neftel et al. 10× single-cell RNA-seq dataset[26] (#GSE131928, GEO—https://www.ncbi.nlm.nih.gov/geo/query/acc.cgi?acc=GSE131928). Count data were downloaded from GEO, and Seurat was used to generating cell clusters as described above. Cell clusters expressing myeloid cell markers were aggregated and their normalized, log-transformed expression data were used to generate the heatmaps (Supplementary Fig. 6).

**Survival prediction of glioma patients using myeloid cluster signatures.** To assess the correlation between our macrophage subtype signatures and survival in glioma patients, we used publicly available datasets: the Chinese Glioma Genome Atlas (CGGA) (RRID:SCR_018802) (mRNAseq_325, Illumina HiSeq) (http://www.cgga.org.cn/download.jsp) and download the dataset we used here: http://www.cgga.org.cn/download?file=download/20200506/CGGA.mRNAseq_325.RSEM-genes.20200506.txt.zip&type=mRNAseq_325&time=20200506)) and The Cancer Genome Atlas (TCGA) GBM dataset (RRID:SCR_003193) (http://www.linkedomics.org/data_download/TCGA-GBM/, and to download the dataset we used, please copy/paste the following URL: http://linkedomics.org/data_download/TCGA-GBM/Human__TCGA_GBM__UNC__RNAseq__GA_RNA__01_28_2016__BI__Gene__Firehose_RSEM_log2.cct.gz)[70]. We analyzed the RNA-seq (GA, Gene level) dataset with 528 samples to use a large dataset for our analysis. For each patient, a signature score was calculated per myeloid cluster signature genes (top 50 DE genes per cluster, DE genes were generated with Seurat::FindAllMarkers() function. Top DE genes = 50 for all clusters except MC02 which only had 14 significant DE genes). Signature scores were then generated using the "score" function from JLaffy/scrabble R package[26] (see details above) which assigns a signature score for each cluster signature per patient. Survival analyses were done using the survival V3.2-7 (RRID:SCR_021137) and survminer V0.4.9 (RRID:SCR_021094) packages. Since signature scores are centered, patient cohorts were stratified into two groups based on the sign of the signature score (above zero = "enriched", below zero = "not enriched"), and the statistical significance of

the difference in clinical outcome was calculated using the log-rank Mantel-Cox test. The survival characteristics of the groups were visualized using Kaplan-Meier curves. Multivariate Cox regression analysis was performed using the survival::coxph() function using the variables specified in the text.

**Cell–cell communication analysis using CellPhoneDB.** We applied an established method CellPhoneDB[38] package V2.1 (RRID:SCR_017054) to study cell-cell interactions across Glioma, Myeloid and T-cell types. CellPhoneDB uses several ligand-receptor databases like IUPHAR, UniProt, Ensemble, and PDB as a reference to evaluate the cellular communication networks between two cell types. We only considered those ligands and receptors that are expressed in at least 10% (default cutoff) of the single cells in a specific cluster. CellPhoneDB performs a pairwise comparison between all the cell types by randomly permuting labels of the clusters 1000 times (default) and determining the mean average expression levels of LR in the given interacting cluster pairs. Finally, CellPhoneDB computes a p-value by calculating the proportion of the means that are equal to or higher than the actual mean for a specific ligand-receptor pair. For plotting, we only considered LR pairs having p-value ≤ 0.05 and mean value >2 of the individual LR partner average expression in the corresponding cell type pairs.

**Primary mouse glioma tumorsphere lines.** Primary glioma tumorsphere lines were established from spontaneous GBMs that formed in the *S100ß-vERBb;p53* mouse model. Briefly, glioma regions were micro-dissected under a dissecting microscope and single-cell suspensions were cultured in neural stem cell medium (DMEM/F12 (HyClone, #Sh30261.01), B27 (#17504-044; Life Technologies), pen/strep (#30002CI; Corning), 10 ng/ml bFGF (#100-18B; PeproTech, Ricky Hill, NJ), and 20 ng/ml EGF(#315-09, PeproTech). The two primary tumorsphere lines used were 5459 (*S100ßverbB;p53+/−*, male) and 2808 (*S100ßverbB;p53−/−*, female).

**In vivo testing of S100a4 function in stromal cells.** Freshly dissociated S100ß-vErbB;p53 tumorsphere cells were injected into the striatum of 6–8-week-old female and male C57BL6/J (IMSR Cat# JAX: 000664, RRID:IMSR_JAX:000664) or S100a4−/− (IMSR Cat# JAX:012904, RRID:IMSR_JAX:012904) syngeneic mice using a stereotaxic device (bregma: 2.8/−0.5/−3.5). The number of mice used in each experiment is indicated in Figs. 6 and 7 and Supplementary Fig. 9. Mice were euthanized using $CO_2$ inhalation when they displayed signs of brain tumors, experienced more than 20% body weight loss, have a BCS (body condition score) of 2 or less, have continuous seizures or other complications associated with hindlimb paralysis. Whole brains were cut into 2 mm coronal sections using a brain mold, and glioma regions were microdissected under a dissecting microscope for analysis. Mice were housed in the HMRI vivarium, which is an AAALAC accredited facility in compliance with the Guide for the Care and Use of Laboratory Animals (Protocol # AUP-0120-0003). Mice have been housed in individually ventilated cages, 4–5 mice per cage. The room environment was maintained at 68–72°F (20–22°), with 30% to 70% humidity, on a 12:12 light:dark cycle. All procedures were approved by the HMRI Animal Care and Usage Committee. B6 or *S100a4−/−* animals were randomly selected for this study and were age- and sex-matched at the time of the injections.

**Immunophenotyping by flow cytometry.** Freshly dissected mouse glioma tissues were microdissected into small chunks and then treated with Accutase for 10–15 min at 37 °C. Accutase was removed, and tissues were resuspended in DME/F12+ B27+ pen/strep medium to generate single-cell suspensions. Cells were resuspended in RBC lysis buffer to remove red blood cells. Following RBC lysis, cells were strained through 40 μm Flowmi cell strainers (#H136800040; Bel-Art, Wayne, NJ). Cells were then stained with multiple flow cytometry validated antibody cocktails (see below) and analyzed using either BD Fortessa or LSRII cytometers. Data were collected using BD FACSDiva Software V9.0 (RRID:SCR_001456) and analyzed/quantified using FlowJo V10.8.0 (RRID:SCR_008520). All antibody dilutions and staining were performed in Brilliant Stain Buffer (#563794, BD Biosciences), and cells were incubated with a blocking solution containing mouse TruStain FcX (#101319; BioLegend, San Deigo, CA) before antibody staining. Antibodies used: PE-cy7 CD45 (#103114, RRID:AB_312979; BioLegend—1:1000), APC-cy7 CD45 (#103116, RRID:AB_312981; BioLegend—1:1000), BV650 CD11b (#101259, RRID:AB_2566568; BioLegend—1:1000), PE CD3e (#100308, RRID:AB_312673; BioLegend—1:1000), BV650 CD4 (#100469, RRID:AB_2783035; BioLegend—1:1000), BV711 CD8a (#100747, RRID:AB_11219594; BioLegend—1:1000), BV711 Ly6C (#128037, RRID:AB_2562630; BioLegend—1:1000), and APC-cy7 Ly6G (#127624, RRID:AB_10640819; BioLegend—1:1000).

**Immunofluorescence analysis.** Tissues were fixed in 4% paraformaldehyde (PFA) overnight, equilibrated through 10, 20, and 30% sucrose gradients, and then embedded in OCT compound (#23-730-571; Thermo Fisher Scientific). Frozen samples were sectioned to 10 μm thickness, and slides were blocked in 5% normal goat serum/0.2% Triton PBS for 30 min and incubated with primary antibodies overnight at 4 °C. Then, appropriate Alexa Fluor secondary antibodies (Invitrogen) were incubated for 30–45 min. Nuclei were stained with DAPI (1:2000, Invitrogen). TrueVIEW autofluorescence quenching kit was applied (#SP-8500-15; Vector

Laboratories, Peterborough, UK) to remove background fluorescence. Images were obtained using the Zeiss Axiovert 200 M fluorescence microscope and the FV3000 confocal microscope (Olympus, Tokyo, Japan). Primary antibodies used: CD3 (#14-0032-85, RRID:AB_467054; Thermo Fisher Scientific—1:2000), S100A4 (#PA5-16586, RRID:AB_10977371; Thermo Fisher Scientific—1:200), CD45 (#CBL1326, RRID:AB_2174425; MilliporeSigma, Burlington, MA—1:200). hCD206 (#MCA2235GA, RRID: AB_322613; Bio-Rad, Hercules, CA, USA— 1:100), hS100A4(#SAB2500902, RRID: AB_10604809; MilliporeSigma, Burlington, MA—1:100), GFP (#AHP975, RRID: AB_566990; Bio-Rad, Hercules, CA, USA— 1:200), FOXP3 (#No. 320001, RRID: AB_439745; Biolegend, San Diego, CA, USA —1:100), CD3(#MA1-90582, RRID:AB_1956722; Thermo Fisher scientific, Waltham, MA,USA- 1:200), CD163(#16646-1-AP, RRID: AB_2756528; Proteintech, Rosemont, IL, USA—1:100), mCD206(#18704-1-AP, RRID: AB_10597232; Proteintech, Rosemont, IL, USA—1:200) and mCD25(#No.101902, RRID: AB_312845; Biolegend, San Diego, CA, USA—1:200). Secondary antibodies: anti-Goat IgG Alexa Flour488 (# A-11055, RRID:AB_2534102; Thermo Fisher Scientific). Anti-Rabbit IgG Alexa Flour488 (# A-11070, RRID:AB_142134; Thermo Fisher Scientific). Anti-Rabbit IgG Alexa Flour594 (# A-11072, RRID:AB_142057;Thermo Fisher Scientific). Anti-Rat IgG Alexa Flour594 (# A-11007, RRID:AB_10561522;Thermo Fisher Scientific).

**Immunohistochemistry analysis**. Tissues were fixed in 10% formalin and embedded in paraffin. Paraffin blocks were sectioned to 5 μm thickness, and deparaffinized and boiled in 10 mM sodium citrate (pH 6) buffer to retrieve antigens. Slides were blocked in 5% goat serum (G9023, SIGMA) for 30 min and incubated with S100A4 primary antibody (#13018 RRID:AB_2750896, Cell Signaling Tech—1:200) overnight at 4 °C. Then, slides were washed and incubated with anti-mouse/rabbit/goat IgG-biotinylated secondary antibody (# BA-1300, RRID:AB_2336188; Vector Laboratories) for 30–60 min, followed by ABC (Vector Laboratories, Cat#PK-8200) for 1 h and DAB (Vector Laboratories) chemogen reaction. Nuclei were counter-stained with Hematoxylin (#95057-844, VMR, US). Images were captured using the Olympus BX 41 microscope.

**T cells proliferation assay**. Naïve B6 splenocytes were isolated from freshly dissected spleens and enriched for CD3+ cells using the Pan T Cell Isolation Kit II (#130-095-130, Miltenyi) and stained with CellTrace™ Far Red Cell (#C34572, Invitrogen™) according to manufacturer's protocols. Cells were then stimulated using CD3/Cd28 dynabeads (#11456D, Thermofisher) and cocultured for 4 days with FACS sorted GFP+ (endogenous) CD3+, (#100308, RRID:AB_312673; BioLegend—1:1000) and CD4+ (#100469, RRID:AB_2783035; BioLegend—1:1000) tumor-infiltrating lymphocytes isolated from S100a4$^{-/-}$ and S100a4$^{+/-}$ hosts injected with x2808 mouse glioma. At day 4 of co-culture cells were analyzed using either BD Fortessa or LSRII cytometers.

Conditioning media from the above experiment were saved and IFNG levels were measured using the ELISA MAX™ Deluxe Set Mouse IFN-γ kit (#430804, Biolegend) according to the manufacturer's protocol.

**Phagocytosis assay**. Tumor-infiltrating GAMs were isolated from B6, S100a4$^{-/-}$ and S100a4$^{+/-}$ hosts injected with ×2808 mouse glioma using FACS sorting of CD45$^{hi}$ (#103114, RRID:AB_312979; BioLegend—1:1000), CD11b$^{hi}$(#101259, RRID:AB_2566568; BioLegend—1:1000) and GFP+ (in case of S100a4$^{-/-}$ vs. S100a4$^{+/-}$). GAMs were then cultured overnight and incubated with pHrodo™ Red Zymosan Bioparticles™ Conjugate for Phagocytosis (P35364, Thermo Fisher Scientific) for two hours on the following day. Following two-hour incubation with the beads, immunofluorescence images were obtained using the Zeiss Axiovert 200 M fluorescence microscope.

**Statistical analysis**. Statistical comparisons were performed using GraphPad Prism V9.3.0 (RRID:SCR_002798; GraphPad Software, La Jolla, CA) or R. Values and error bars represent the mean ± standard error of the mean. The respective number of replicates (n) is indicated in the figures or in the figure legends. Power analyses were used to determine appropriate sample sizes for animal experiments (power 0.8, alpha 0.05). p-Values were determined by an appropriate statistical test such as Student's t-test or analysis of variance (ANOVA) with multiple comparison correction, as indicated in the figure legends.

**Reporting summary**. Further information on research design is available in the Nature Research Reporting Summary linked to this article.

## Data availability
The raw single-cell sequencing data generated in this study are publicly available with no restrictions through the GEO series GSE182109. The raw exome sequencing data generated in this study have been deposited in SRA under the accession code PRJNA787981. The Neftel et al. [26] The 10× single-cell RNA-seq publicly available data used in this study is available through GEO #GSE131928. The publicly available CGGA (Chinese Glioma Genome Atlas) dataset used in this study (mNRAseq_325) is available through the following link: http://www.cgga.org.cn/download.jsp and downloadable from http://www.cgga.org.cn/download?file=download/20200506/CGGA.mRNAseq_ 325.RSEM-genes.20200506.txt.zip&type=mRNAseq_325&time=20200506. The publicly available The Cancer Genome Atlas (TCGA) GBM dataset used in this study is available through this link: http://linkedomics.org/data_download/TCGA-GBM/ and downloadable using the following link: http://linkedomics.org/data_download/TCGA-GBM/Human__TCGA_GBM__UNC__RNAseq__GA_RNA__01_28_2016__BI__Gene__Firehose_RSEM_log2.cct.gz. Source data are provided with this paper. The remaining data are available within the Article, Supplementary Information, or Source Data files. Source data are provided with this paper.

## Code availability
All codes to reproduce figures presented in this paper will be publicly available through GitHub (https://github.com/parveendabas/GBMatlas) and the corresponding DOI is as follows: https://doi.org/10.5281/zenodo.5765535.

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

## Acknowledgements
We thank Tetsuo Ashizawa, Dorothy Lewis, Brandi Mattson, and Taneli Helenius for their editorial and Matthew G. Landry for graphic design assistance. We also thank Ping Li for her technical assistance. This study was supported by grants from the NIH (1R01NS121405, K.Y.), Cancer Prevention & Research Institute of Texas (RP180882, K.Y.), Department of Defense (W81XWH-14-1-0115, K.Y.), the Houston Methodist Foundation (K.Y.), the Donaldson Charitable Foundation (K.Y.), The Peak Foundation (D.S.B.), and the DOD Horizon Award (CA191052: NA). This study was supported by The University of Texas MD Anderson Moon Shots Program™ (F.F.L.), the National Cancer Institute P50CA127001 (F.F.L.), The Broach Foundation for Brain Cancer Research (F.F.L.), and The Elias Family Fund (F.F.L.).

## Author contributions
K.Y. conceived the study, designed experiments, and analyzed the data. N.A., J.S.L., and C.W. designed experiments and collected and analyzed the data. P.K., J.G., R.G., W.F.F., and N.A. analyzed the sequencing data. D.S.B., O.B.I., K.P., S.W., B.Y.S.K., F.F.L., K.H., S.S.P. provided the clinical samples. B.Y.S.K., W.J., S.Z.P., and D.L.H. analyzed the data. All authors contributed to preparing the paper.

## Competing interests
K.Y. is a co-founder of EMPIRI, Inc. The remaining authors declare no competing interests.
