## [Peer Review File · Nature Communications]

Single-cell analysis of human glioma and immune cell
sidentifies S100A4 as an immunotherapy targetReviewers' Comments:

Reviewer #1:

Remarks to the Author:

This is an interesting manuscript. There have been a large number of single cell studies in brain tumors with a smaller number of studies of immune cell studies using either scRNAseq or CyTOF, including studies that were not referenced by the authors:

1. Immune profiling of human tumors identifies CD73 as a combinatorial target in glioblastoma. Goswami S, Walle T, Cornish AE, Basu S, Anandhan S, Fernandez I, Vence L, Blando J, Zhao H, Yadav SS, Ott M, Kong LY, Heimberger AB, de Groot J, Sepesi B, Overman M, Kopetz S, Allison JP, Pe'er D, Sharma P. *Nat Med.* 2020 Jan;26(1):39-46. doi: 10.1038/s41591-019-0694-x.
2. Distinct regional ontogeny and activation of tumor associated macrophages in human glioblastoma. Landry AP, Balas M, Alli S, Spears J, Zador Z. *Sci Rep.* 2020 Nov 11;10(1):19542. doi: 10.1038/s41598-020-76657-3.
3. Single-Cell Atlas Reveals Complexity of the Immunosuppressive Microenvironment of Initial and Recurrent Glioblastoma. Fu W, Wang W, Li H, Jiao Y, Huo R, Yan Z, Wang J, Wang S, Wang J, Chen D, Cao Y, Zhao J. *Front Immunol.* 2020 May 7;11:835. doi: 10.3389/fimmu.2020.00835.

The current study has a limited number of tumors, but contributes to the literature. There are some concerns beyond novelty.

Major concerns:

1. The multi-region biopsies were not used to their full potential, like Landry et al. The biopsies should be mapped to specific MRI features to address the regional variation. Figure 1b shows that clustering was variable between regions and patients. It would be useful to have a consistent consideration of the different regions. It is common for necrotic regions to have more macrophage and this should be explicitly addressed.
2. I do not understand the use of Neftel et al. for this study as the focus here is on macrophage, which were excluded from the Neftel study.
3. Please provide more details on the validation cohort of 9 patients. Is this an existing dataset or newly derived?
4. I am concerned about the in silico survival studies. The cut-off rules were not clearly specified and I cannot replicate the S100A4 results from the authors. Please insure that all analyses use an a priori cut-off, like median expression, instead of p-hacking for optimized p-values. Please list the number of samples in each subgroup.
5. The results with the S100a4^{-/-} mice are interesting, but do not directly address S100A4 as an immune target. The studies are quite preliminary. The immune changes seen are purely correlative and not functional. As S100A4 is widely expressed, it cannot be assumed that the cause is solely or directly through immune responses.
6. It would be useful to include direct validation of some of the targets, especially S100A4, in regional staining of patient specimens by immunohistochemistry or immunofluorescence.

Minor concerns:

1. The figure call outs are sometimes in error. The size bar for figure 4e is missing.
2. The text itself could have improved quality of the writing and grammar.
3. Statistical testing should be more detailed.
4. Please better indicate the number of replicates in each figure.

Reviewer #3:

Remarks to the Author:

Abdelfattah et al., present new data that can possibly lead towards new immune therapy based interventions in Glioblastoma.

immunotherapy has not been effective against glioblastoma (GBM). A lot of progress has been made recently towards understanding GBM tumor heterogeneity and its association with poor outcomes, particularly by applying single cell genomic approaches. However, the immune landscape has not been characterized at the same level of detail and Abdelfattah et al aim to address this. This is a novel angle and of interest to the wider brain tumour research community.

The authors performed scRNA-seq using 16 samples derived from four patients (three grade IV GBMs and one LGG, all IDH wt), four from each tumor. This is a good number of samples to cover intratumor heterogeneity and represent stroma cells, however it is not a convincing number in terms of intertumor heterogeneity. The number of normal cells in each tumor differed considerably also (see suppl figure 2a). So, with these tumor numbers I cannot agree with the authors that they established the first comprehensive cell atlas for human glioma.

Nevertheless, 60,024 single cells were analyzed for scRNA-seq expression-based copy number analysis. The authors used their own tool COPYKAT. This work is still in review apparently, so a more detailed description should be included.

The authors concentrated on the identification of immune-suppressive cell types, and the analysis of myeloid cells amongst 33,962 cells. This part of the work is quite detailed leading to the identification of 13 clusters. Cluster signature analysis lead to the identification of immune-suppressive macrophages associated with patterns of high proliferation. Overall the authors identified pro-inflammatory microglia and both pro-inflammatory and immune suppressive bone marrow derived macrophages.

The signatures of these were associated with outcome in the Chinese Glioma Genome Atlas and TCGA datasets. The authors were also able to identify the macrophage subpopulations in an independent GBM dataset by Neftel et al. and similar clustering is reported.

There is no validation of the key signature genes in patient tumor tissues.

One of the markers for the suppressive macrophages as well as Tregs is S100A4. Knock out mice for this gene have reported defects in macrophage infiltration. The authors used this mouse model to demonstrate that in these mice the tumors are not infiltrated by the suppressive macrophage population and survived after tumorshpere glioma cell line transplantation much longer than wildtype mice. Immune cells isolated from these (knock out mouse) glioma tumors show enhanced T-cell infiltration and higher ratios of T-cells to myeloid cell ratios. Indeed, this does show that S100A4 represents a novel target and possible new avenue for an effective immunotherapy approach.

Major points to address:

1. No validation of key markers in patient tumor tissue
2. Number of patients is modest for an Atlas type publication
3. The authors miss the opportunity to perform tumour-host interaction analysis in the patient tumours, map interactions and demonstrate the cross-talk between GBM cells and myeloid cells, as well as other immune cells (T-cells etc). This would significantly strengthen the work.

Minor points:

One of the phenotypes of the S100A4^{-/-} mice is spontaneous tumor formation (Naaman et al 2004).

So, one can assume that the proposed inhibitors would need to be cell type specific. The cancer predisposition of this model and possible implications is worth including in the discussion.

There is more recent work by the Bresnick group on the macrophage defect of the knock out mice than reference 40, which can be cited.

Reviewer #4:

Remarks to the Author:

the manuscript titled "A glioma immune cell atlas identifies prognostic macrophage signatures and a novel immunotherapy target, S100A4", Abdelfattah and Kumar et al. attempts to utilize single-cell analysis of glioma samples obtained from 4 patients to define molecular signatures of inflammatory and immunosuppressive macrophages. Using their single-cell sequencing data, the authors identify S100A4 as a potential target found in immune-suppressing subtypes of macrophages and of the lymphoid lineage- regulatory T cells. By comparing growth of orthotopically transplanted syngeneic glioma tumorsphere cell lines in wildtype and S100a4^{-/-} hosts, they show deletion of S100A4 can inhibit tumor growth and can promote infiltration of CD4 and CD8 T cells into the tumor. Overall, the detailed profiling of macrophage subpopulations in glioma conducted by the authors has the potential to help identify additional candidates that can be targeted to modulate myeloid populations in GBM, which often constitute the majority of immune cells in brain tumors. However, the small sample size limits the study's conclusions and generalizability.

Major comments:

1. While the authors' conclusions are reasonable, the conclusions are significantly limited by the small sample size of the study. While the authors obtained multiple samples from each patient, ultimately, the samples only represented two wild type GBM, one recurrent GBM, and one grade II tumor. As prior studies have concluded, the tumor microenvironment and transcriptomics of treatment naïve GBM and recurrent GBM post chemoradiation are significantly different. Therefore, I have significant concerns in forming conclusions when combining these biologically different samples. This is also evident by the fact that a majority of the myeloid clusters identified by the authors are contributed by only 1-2 patient samples, with the recurrent GBM patient contributing to the vast majority of cells in majority of the clusters raising the question of the generalizability of the conclusions. Furthermore, while on a single cell level, the different GBM meta-modules are represented, the small sample size does not represent the significant heterogeneity of these tumors.
2. While the authors use GSEA to determine the activation scores of different hallmark enrichment pathways to determine whether the myeloid cluster are immunosuppressive or pro-inflammatory, though it is known that many of these pathways can be activated in either condition of myeloid cells. The authors should first validate the presence of these myeloid clusters by protein expression to validate their existence, and further validate the immunosuppressive functions of these myeloid clusters with in vitro assays to support their conclusions.
3. The authors evaluated whether the signature of each myeloid clusters were associated with GBM survival in bulk publicly available datasets, however they do not discuss whether they account for other factors that are known to stratify patient survival such as IDH mutation status or MGMT status in performing the survival analysis which can significant confound their results.
4. While the authors identified S100A4 as a potential target found in Treg and immunosuppressive myeloid cells. They should also evaluate the expression of S100A4 in other cell types including tumor cells and stromal cells to determine whether the cancer promoting role of S100A4 are mainly contributed by immune cells versus tumor or stromal cells. Along those lines, the knockout study demonstrating the survival benefit and increase in immune cell infiltration in S100A4 knockout model are not specific to knockout of S100A4 in immunosuppressive myeloid cells or Tregs and therefore it is difficult to conclude that the benefit seen is due to decreased expression of S100A4 in those cell types.
5. The authors suggest targeting S100A4 may simultaneously reverse immunosuppressive phenotypes of both Tregs and immunosuppressive myeloid cells. Were depletion studies conducted in S100a4^{-/-}

mice to selectively deplete Tregs or myeloid cells, to better understand the contribution of targeting S100a4 on either/both populations toward an enhanced anti-tumor immunity?

6. S100a4^{-/-} macrophages have compromised chemotaxis in vitro and infiltration to inflamed sites in vivo, but the authors did not see any difference in myeloid infiltration into the tumor of B6 and S100a4^{-/-} mice. A functional assay to compare immunosuppressive activity of S100a4^{-/-} macrophages with that of B6 mice (or even of Tregs) would complement the current manuscript and be helpful in delineating MOA of S100a4^{-/-} ablation/inhibition.

Minor comments:

1. Please identify the figure numbers correctly to correspond with the discussion in the text. For example- line 190 mentions figure 4 but should be referring to figure 3. Some of the supplementary figure numbers are also used incorrectly in the text.
2. There are minor grammatical errors throughout the text, please edit them.
3. Figure 4 and supplementary figure 7 show the difference in tumor-infiltrating CD3+, CD8+ and CD4+ T cells in B6 vs S100a4^{-/-} hosts. Since Tregs also have CD4 expression, were markers of Tregs or helper T cells included to compare Treg vs helper subsets in B6 and S100a4^{-/-} mice?

Rebuttal letter for NCOMMS-20-44885A

We thank the reviewers for their thoughtful comments and suggestions. Please see below for our point-by-point response to their comments. Please note that we have significantly increased the sample number (201,986 cells from 44 samples obtained from 18 patients). As a consequence, the results section was largely re-written to describe the much larger dataset we present now and to accommodate reviewer suggestions. We were able to replicate the major conclusions from our original submission.

Reviewer #1

“This is an interesting manuscript. There have been a large number of single cell studies in brain tumors with a smaller number of studies of immune cell studies using either scRNAseq or CyTOF, including studies that were not referenced by the authors:”

We thank the reviewers for pointing out this oversight. These are now added.

25. Immune profiling of human tumors identifies CD73 as a combinatorial target in glioblastoma. Goswami S, Walle T, Cornish AE, Basu S, Anandhan S, Fernandez I, Vence L, Blando J, Zhao H, Yadav SS, Ott M, Kong LY, Heimberger AB, de Groot J, Sepesi B, Overman M, Kopetz S, Allison JP, Pe'er D, Sharma P. Nat Med. 2020 Jan;26(1):39-46. doi: 10.1038/s41591-019-0694-x.

24. Distinct regional ontogeny and activation of tumor associated macrophages in human glioblastoma. Landry AP, Balas M, Alli S, Spears J, Zador Z. Sci Rep. 2020 Nov 11;10(1):19542. doi: 10.1038/s41598-020-76657-3.

18. Single-Cell Atlas Reveals Complexity of the Immunosuppressive Microenvironment of Initial and Recurrent Glioblastoma. Fu W, Wang W, Li H, Jiao Y, Huo R, Yan Z, Wang J, Wang S, Wang J, Chen D, Cao Y, Zhao J. Front Immunol. 2020 May 7;11:835. doi: 10.3389/fimmu.2020.00835.

Major concerns

1) *“The multi-region biopsies were not used to their full potential...the biopsies should be mapped to specific MRI features to address the regional variation. Figure 1b shows that clustering was variable between regions and patients. It would be useful to have a consistent consideration of the different regions. It is common for necrotic regions to have more macrophage and this should be explicitly addressed”.*

Thank you for the suggestion. We performed unbiased and systemic analysis of all samples from multi-regional sampling to decipher spatial heterogeneity (36 fragments from 10 patients, Fig 5a) and present the results in new Figure 5 and Supplementary Figure 7. This analysis confirmed our original conclusion that there is significant heterogeneity in immune infiltrates in different regions in the same patient.

The clinical annotation was available for 19 of the 44 samples (Supp Table 1); however, only three of the 19 samples were annotated to originate from necrotic regions.

In the revised manuscript, we added the following.

“In our dataset, only three samples were annotated to originate from necrotic regions (**Fig 5b**, red bars), and they did not necessarily contain more macrophages (microglia + BMDMs) than other samples. The samples from invading/infiltrating regions (**Fig 5b**, green bars) did contain more microglia than BMDMs, supporting previous studies^{15,24}.”

In summary, single cell level analysis of intra-and inter-tumoral heterogeneity, using molecularly defined glioma, T cell, microglia and BMDM cell subtypes (**Fig 5 and Supp Fig 7a,b**), demonstrates significant cellular heterogeneity of cancer and immune cells in gliomas.” Page 13, lines 288-294.

2) *“I do not understand the use of Neftel et al. for this study as the focus here is on macrophage, which were excluded from the Neftel study”*

The reviewer is correct that the Neftel *et al.* paper focused on glioma cells only; however, their dataset also included glioma associated immune cells. We reanalyzed their dataset and extracted just the myeloid cells to use as an independent cohort validation of our macrophage signature genes. Since we have a large number of independent patient samples now this analysis less important; however, we present the results from this independent cohort in Supp Fig 6 to illustrate the generalizability of our myeloid subtype signatures.

3) *“Please provide more details on the validation cohort of 9 patients. Is this an existing dataset or newly derived?”*

Please see response above, as the 9 samples in the validation cohort are from Neftel *et al.*

4) *“I am concerned about the in silico survival studies. The cut-off rules were not clearly specified and I cannot replicate the S100A4 results from the authors”.*

We apologize for the difficulty the reviewer had in reproducing our results, and do not understand why that is the case. We are able to consistently generate the same result for S100A4 survival in current **Figure 6d** using the code we deposited and the dataset we identified. Median values were designated as cut-off in all survival analysis in this paper. For your convenience, we have created and exported a CodeOcean capsule containing the necessary input files as well as all required code and packages to generate the results. You can recreate the exact figure independently by extracting and executing this container. <https://hmethodist.box.com/s/j4udtqe57dxinaw2bjstubxj9zeha4ng>.

As for the survival analysis in Figure 4h, i and Supplementary Figure 5b, the cutoff value is zero. The meta-module scores are centered around zero and we consider positive scores as positive association/enrichment and negative scores as negative association/enrichment. However, the algorithm used to generate the meta-module score results in slight variations every time the analysis is performed. Regardless of these small variations, the conclusions we make in the paper remain unchanged (i.e. the trend and significance remain the same). This is clearly annotated within the deposited code with the recommendation to save the generated cell-score for future reproducibility. Figure legends now clearly state that the cut-off is median expression value for Figure 6d and cut-off =0 for meta-module scores in Figure 4h, i and Supplementary Figure 5b.

5) *“The results with the S100a4^{-/-} mice are interesting, but do not directly address S100A4 as an immune target. The studies are quite preliminary. The immune changes seen are purely correlative and not functional. As S100A4 is widely expressed, it cannot be assumed that the cause is solely or directly through immune responses.”*

Thank you for this feedback. Please see new data presented in Figure 7. We show through functional assays that S100a4^{-/-} myeloid cells have significantly increased phagocytic activity (Fig 7b,c) and that S100a4^{-/-} CD4 T cells are more activated, demonstrated by increased T cell proliferation and interferon gamma secretion (Fig 7d,e). Importantly, we performed these

functional assays by specifically isolating S100a4-expressing (GFP+) myeloid and T cells in gliomas to demonstrate the cell-autonomous function of S100a4.

6) *“It would be useful to include direct validation of some of the targets, especially S100A4, in regional staining of patient specimens by immunohistochemistry or immunofluorescence”*

Thank you for this suggestion. We have now added immunohistochemistry and double immunofluorescence results from human and GBM samples (new Fig 6c, Supplementary Figure 8).

Minor concerns:

1. The figure call outs are sometimes in error. The size bar for figure 4e is missing.

We apologize for the mistakes we made while rushing to prepare the original manuscript. These are now corrected in the revised version.

2. The text itself could have improved quality of the writing and grammar.

We apologize for this error. This revised manuscript has been extensively edited.

3. Statistical testing should be more detailed.

We now provide more detailed information on statistical tests used within each legend and in Supplementary Methods. Please see legends for Figures 2, 4, 6, 7 and Supplementary Figure 9.

4. Please better indicate the number of replicates in each figure.

These are now included in the revised figure legends.

Reviewer #3

The authors performed scRNA-seq using 16 samples derived from four patients (three grade IV GBMs and one LGG, all IDH wt), four from each tumor. This is a good number of samples to cover intratumor heterogeneity and represent stroma cells, however it is not a convincing number in terms of intertumor heterogeneity. The number of normal cells in each tumor differed considerably also (see suppl figure 2a). So, with these tumor numbers I cannot agree with the authors that they established the first comprehensive cell atlas for human glioma.

In response to reviewer's comments, we increased our sample size significantly. We now present data from 201,988 cells from 18 patients and 44 fragments.

The authors used their own tool COPYKAT. This work is still in review apparently, so a more detailed description should be included.

This manuscript is now published, and the details are provided in Gao et al., *Nature Biotechnology*, 2021. Ref #21.

Gao, R. et al. Delineating copy number and clonal substructure in human tumors from single-cell transcriptomes. *Nat Biotechnol* **39**, 599-608, doi:10.1038/s41587-020-00795-2 (2021).

Major points to address:

1) *“No validation of key markers in patient tumor tissue”*

Please see our response to reviewer #1 comment 7 above. We added Supplementary Figure 8 to show expression of S100A4 in glioma associated T cells and macrophages, and added Figure 6c to demonstrate co-expression of S100A4 with CD206 and CD163 in human and mouse GBM.

2) *“Number of patients is modest for an Atlas type publication”*

We more than quadrupled the patient number (18 patients) and analyzed 201,988 single cells from 44 fragments in this revised paper.

3) *“The authors miss the opportunity to perform tumour-host interaction analysis in the patient tumours, map interactions and demonstrate the cross-talk between GBM cells and myeloid cells, as well as other immune cells (T-cells etc).”*

Thank you for this suggestion. New Figure 5 and Supplementary Figure 7 are focused on analyzing spatial heterogeneity of cancer and immune infiltrates. We also identify and illustrate shared and unique cell:cell communication among different cancer and immune cells types using the Cellphone DB analysis tool. Results are discussed on page 14-15, line# 304- 326.

Minor concerns.

4) *“One of the phenotypes of the S100A4^{-/-} mice is spontaneous tumor formation (Naaman et al 2004). So, one can assume that the proposed inhibitors would need to be cell type specific. The cancer predisposition of this model and possible implications is worth including in the discussion”*

Thank you for pointing this out. We have maintained S100a4^{-/-} mice in our colony for nearly a decade and have not observed any spontaneous tumor formation. It is not clear what the difference is but it could be differences in genetic background or housing condition. We did observe reduced viability of homozygous embryos when S100a4^{-/-} was crossed to a mixed genetic background strain for a different study. Hence, we back-crossed the original strain we received from Dr. Eric Neilson to C57Bl6/J for several generations and since maintained our colony in the pure C57Bl6/J background.

In response to reviewer comments, we now added “Although an earlier study reported increased tumor formation in S100a4^{-/-} mice⁶⁰, we did not observe spontaneous tumors in our S100a4^{-/-} colony in over a decade.” on page 22, line# 488-489 of Discussion to mention this potential complication.

5) *“There is more recent work by the Bresnick group on the macrophage defect of the knock out mice than reference 40, which can be cited”*

Apologies for this omission. We now added Dulyaninova et al., 2018 paper as reference #45.

45. Dulyaninova, N. G., Ruiz, P. D., Gamble, M. J., Backer, J. M. & Bresnick, A. R. S100A4 regulates macrophage invasion by distinct myosin-dependent and myosin-independent mechanisms. *Mol Biol Cell* **29**, 632-642, doi:10.1091/mbc.E17-07-0460 (2018).

Reviewer #4

Major comments:

1) *“While the authors’ conclusions are reasonable, the conclusions are significantly limited by the small sample size of the study. While the authors obtained multiple samples from each patient, ultimately, the samples only represented two wild type GBM, one recurrent GBM, and one grade II tumor. As prior studies have concluded, the tumor microenvironment and transcriptomics of treatment naïve GBM and recurrent GBM post chemoradiation are*

significantly different. Therefore, I have significant concerns in forming conclusions when combining these biologically different samples. This is also evident by the fact that a majority of the myeloid clusters identified by the authors are contributed by only 1-2 patient samples, with the recurrent GBM patient contributing to the vast majority of cells in majority of the clusters raising the question of the generalizability of the conclusions. Furthermore, while on a single cell level, the different GBM meta-modules are represented, the small sample size does not represent the significant heterogeneity of these tumors”

In response to reviewer comments, we have increased our sample number to 18 patients and 44 fragments, including two grade II astrocytoma, one high grade astrocytoma (grade III), one grade II oligodendroglioma, nine primary GBM, and five recurrent GBM patients (see Supplementary Table 1). We now present analyses from 201,988 total cells from 44 samples, of which 83,479 myeloid cells. De novo clustering and analysis of myeloid cells from this larger dataset show that all patients contribute to myeloid cell clusters (Fig 4c & Supplementary Figure 4b, c), and that there is not a statistically significant difference between ndGBM and rGBM in terms of BMDM/macrophage and MDSC infiltration, although the microglia are significantly reduced in rGBM (see figure). Importantly, our key conclusions held true even when our sample size more than tripled.

2) “While the authors use GSEA to determine the activation scores of different hallmark enrichment pathways to determine whether the myeloid cluster are immunosuppressive or pro-inflammatory, though it is known that many of these pathways can be activated in either condition of myeloid cells. The authors should first validate the presence of these myeloid clusters by protein expression to validate their existence, and further validate the immunosuppressive functions of these myeloid clusters with in vitro assays to support their conclusions”

Thank you for this suggestion. We have now added IHC and IF data from GBM patients and mouse gliomas showing S100A4 expression in immune suppressive macrophages (CD206/CD163+) and FOXP3+ T cells (Fig 6c, Supplementary Figure 8). We also show functional data that glioma associated S100a4+ macrophages have compromised phagocytic function and when S100a4 is deleted from these cells, glioma associated macrophages show significantly increased phagocytic activity (Fig 7b,c).

3) “The authors evaluated whether the signature of each myeloid clusters were associated with GBM survival in bulk publicly available datasets, however they do not discuss whether they account for other factors that are known to stratify patient survival such as IDH mutation status or MGMT status in performing the survival analysis which can significant confound their results.”

We apologize for this oversight. It now explicitly states on page 11 and 12 that we performed multivariate Cox regression analysis with gender, tumor type, IDH status, and MGMT status as potential variables and report that myeloid subtype gene signature are independent predictors of good or poor survival (5 of 9 signatures for all gliomas and 3 of 9 for GBMs, Supplementary Table 10).

4) *“While the authors identified S100A4 as a potential target found in Treg and immunosuppressive myeloid cells. They should also evaluate the expression of S100A4 in other cell types including tumor cells and stromal cells to determine whether the cancer promoting role of S100A4 are mainly contributed by immune cells versus tumor or stromal cells. Along those lines, the knockout study demonstrating the survival benefit and increase in immune cell infiltration in S100A4 knockout model are not specific to knockout of S100A4 in immunosuppressive myeloid cells or Tregs and therefore it is difficult to conclude that the benefit seen is due to decreased expression of S100A4 in those cell types”*

We previously published that S100A4 is a marker and a regulator of glioma stem cells (GSCs) in human and mouse gliomas (Chow et al., 2017 Cancer Research, ref# 43). So the reviewer is correct that S100a4 expression in glioma cells also contribute to glioma growth. This manuscript was referenced in the original submission but now we also add Figure 6c and Supplementary Figure 8 showing S100A4 expression in T regs and CD163+ and CD206+ macrophages in human and mouse. We also reported S100A4 expression in perivascular GSCs in the same paper, but it is formally possible that some of these cells are pericytes (although the majority are GSCs). We do not observed S100A4 expression in endothelial cells.

We would like to clarify that the S100a4^{-/-} host study was performed with unmanipulated glioma cells to specifically address whether S100a4 expression in stromal cells affect glioma growth. As shown in Figure 6f, S100a4^{+/+}GFP⁺ glioma cells are present in S100a4^{-/-} host gliomas. In addition, S100A4-GFP⁺ CD206⁺ macrophage and FOXP3⁺ T cells are also present, indicating that S100a4 deletion did not block their migration or lineage determination (Fig 6f). Others have reported that S100A4 is expressed in a subset of astrocytes but absent from microglia in the normal brain. S100A4 expression increase upon injury, In both reactive astrocytes and some microglia. However we observed that S100A4 expression remains low in all glioma associated microglia except slight increase in MC6 (Fig 6a), which is the only microglia cluster associated with worse survival (Fig 4h). Therefore, we cannot rule out the potential contribution of mutant astrocytes. However, our functional assays were performed with S100a4/GFP⁺ FACS sorted macrophage and CD4 T cells (Fig 7) to demonstrate that S100a4 function in these immune cells are critical for immune suppression.

5) *“The authors suggest targeting S100A4 may simultaneously reverse immunosuppressive phenotypes of both Tregs and immunosuppressive myeloid cells. Were depletion studies conducted in S100a4^{-/-} mice to selectively deplete Tregs or myeloid cells, to better understand the contribution of targeting S100a4 on either/both populations toward an enhanced anti-tumor immunity”*

In response to multiple reviewer comments, we performed functional assays in vitro to address this concern. We isolated S100a4-expressing (using the GFP reporter that was knocked into the S100a4 locus, which we previously published to coincide 100% with endogenous S100A4 expression (Chow et al., 2017, ref# 43) to determine the functional consequence of S100a4 deletion specifically in S100a4-expressing myeloid and CD4⁺ T cells. By isolating GFP⁺CD11b⁺ or GFP⁺CD3⁺CD4⁺ cells to perform functional assay, we provide direct evidence for cell-autonomous functions of S100a4 in these cell types.

6) *“S100a4^{-/-} macrophages have compromised chemotaxis in vitro and infiltration to inflamed sites in vivo, but the authors did not see any difference in myeloid infiltration into the tumor of B6 and S100a4^{-/-} mice. A functional assay to compare immunosuppressive activity of S100a4^{-/-} macrophages with that of B6 mice (or even of Tregs) would complement the current manuscript and be helpful in delineating MOA of S100a4^{-/-} ablation/inhibition”*

Thank you for the suggestion. Please see our response above.

Minor comments:

1) *Please identify the figure numbers correctly to correspond with the discussion in the text*

We apologize for this oversight. We confirmed that new figures are correctly identified in the revised manuscript.

2) *There are minor grammatical errors throughout the text, please edit them.*

We again apologize for this mistake from rushing the manuscript preparation. This revised manuscript has been thoroughly edited for grammatical errors.

3) *Figure 4 and supplementary figure 7 show the difference in tumor-infiltrating CD3+, CD8+ and CD4+ T cells in B6 vs S100a4-/- hosts. Since Tregs also have CD4 expression, were markers of Tregs or helper T cells included to compare Treg vs helper subsets in B6 and S100a4-/- mice?*

Thank you for pointing this out. Our flow cytometry analysis did not include definitive Treg markers such as FOXP3 since the number of Tregs in mouse gliomas is **very low** and we were not able to detect them consistently. We show in Fig 6b and c that S100A4 expression is the highest in TC4 and TC5 (Tregs and exhausted CD4+ T cells) and co-expression of S100A4 and FOXP3 in a mouse glioma. In addition, we performed functional assays with FACS sorted GFP+CD3+CD4+ T cells from control and S100a4-/- mice (Fig 7d, e). We now revised our text to read “We surveyed signature genes in MC3, MC4, and MC5 pro-tumorigenic myeloid cells and TC4 (Tregs) and TC5 (exhausted T cells, PD1+, CTLA4+, and LAG3+) to discover highly-expressed genes that are shared and may be manipulated to reprogram both innate and adaptive immune cells in GBM.” On page 15.

Reviewers' Comments:

Reviewer #3:

Remarks to the Author:

The authors have responded positively to the review and substantially increased the number of tumors and cases analyzed to 44 and 18 respectively. This represents now an extensive dataset.

All the points raised were addressed, the manuscript now includes a substantial section of spatial heterogeneity and cell:cell interaction data, including PTPRZ1/PTN and SPP1/CD44 signalling (eg figure 5. and suppl. fig.7), which provided material for an enhanced discussion.

Further points regarding additional references and validation experiments have been adequately addressed, including a new supplier. figure 8.

Reviewer #4:

Remarks to the Author:

Comments have been addressed. No further concerns.

Reviewer #5:

Remarks to the Author:

I have a fundamental problem with this paper and it is the combination of biologically different tumor entities, a point that has been raised by reviewer 4 as well. This problem hasn't been adequately addressed at revision.

In particular, the authors lump together IDH mutant and IDH wt tumors, 1p;19q co-deleted and non co-deleted tumors, which are different entities. Also, the diagnosis of the samples listed in the text (lines 96,97) are different from those listed in the rebuttal (page 5) and in some cases not supported by the combination of mutations listed in Table S1. Case 11 cannot be a diffuse astrocytoma if it is IDH mutant and 1p;19q co-deleted, it is an oligodendroglioma. Case 3 and case 10 seem to be IDH wt (assessed how? Only immunohistochemistry for the most common R132H mutation or sequencing?); if these are confirmed IDH wt then they are most likely molecular IDH wt GBM (ie those cases where the surgical sample does not contain the histological hallmarks of GBM but can nowadays be identified as such molecularly). The only way this problem can be addressed is by removing the 2 oligodendroglioma cases and also case 3 and 10, if their diagnosis cannot be ascertained with confidence and re-analyse the data. All conclusions drawn will then be relevant for GBM IDH wt and differences could be explored between primary and recurrence tumors. However, IDH mutant and IDH wt glial tumors are different biological entities and cannot be analysed together, nor conclusions such as ".....suggesting increased T cell infiltration during glioma progression" (line 168) or ".....suggesting increased Tregs in GBM compared to LGG" (line 178) or ".....and lower microglia in GBM compared to LGG" (line 215) etc. can be drawn from this dataset.

Minor point, the Suva group has now published a new manuscript where they analyse also the immune cells, Hara et al Cancer Cell 2021, it would seem appropriate to cite this work and comment on how their conclusions relate the those of this manuscript.

Rebuttal letter for NCOMMS-20-44885A

We thank the reviewers #3 and #4 for accepting this manuscript. We also thank reviewer #5 for thoughtful comments and below are our responses.

1) I have a fundamental problem with this paper and it is the combination of biologically different tumor entities, a point that has been raised by reviewer 4 as well. This problem hasn't been adequately addressed at revision.

Please note that Reviewer#4 had requested that we consider IDH and MGMT status as cofounding factors for survival analyses, and we had successfully addressed this to reviewer #4's satisfaction on the last revision.

2) In particular, the authors lump together IDH mutant and IDH wt tumors, 1p;19q co-deleted and non co-deleted tumors, which are different entities. Also, the diagnosis of the samples listed in the text (lines 96,97) are different from those listed in the rebuttal (page 5) and in some cases not supported by the combination of mutations listed in Table S1. Case 11 cannot be a diffuse astrocytoma if it is IDH mutant and 1p;19q co-deleted, it is an oligodendroglioma. Case 3 and case 10 seem to be IDH wt (assessed how? Only immunohistochemistry for the most common R132H mutation or sequencing?); if these are confirmed IDH wt then they are most likely molecular IDH wt GBM (i.e. those cases where the surgical sample does not contain the histological hallmarks of GBM but can nowadays be identified as such molecularly). The only way this problem can be addressed is by removing the 2 oligodendroglioma cases and also case 3 and 10, if their diagnosis cannot be ascertained with confidence and re-analyse the data. All conclusions drawn will then be relevant for GBM IDH wt and differences could be explored between primary and recurrence tumors. However, IDH mutant and IDH wt glial tumors are different biological entities and cannot be analysed together, nor conclusions such as ".....suggesting increased T cell infiltration during glioma progression" (line 168) or ".....suggesting increased Tregs in GBM compared to LGG" (line 178) or ".....and lower microglia in GBM compared to LGG" (line 215) etc. can be drawn from this dataset.

We thank the reviewer #5 for pointing out potential mis-diagnoses of the three samples. We double checked the pathology reports for these sample and updated the following.

- case 11 (LGG-03): there was an error in copy/pasting the mutational profile for sample 11. It had Atrx loss instead of 1p;19q co-deletion and we corrected this error in the revised Supplementary Table 1. The final diagnosis of this sample remained a diffuse astrocytoma.

- case 3 (previous LGG-02): there was an amended pathology report indicating that LGG-2 was indeed a GBM. We now renamed this sample ndGBM10. We also amended our figures, tables and analysis to regroup this sample with ndGBMs instead of LGGs

- case 10 (ndGBM-08) was correctly identified as GBM.

IDH status was determined by sequencing as well as IHC in all IDH wt samples and we are confident about the diagnosis of these samples.

The primary goal of this study was to analyze the cellular heterogeneity of all glioma and immune cells as the title suggests. Per reviewer's comment, we removed LGG comparison and only compared IDH wt newly diagnosed and recurrent GBM on line 167 (previous line 168). It

now reads: “T and NK cells represented $6.8\pm 2.7\%$ of ndGBMs, and $14.3\pm 8.9\%$ of rGBMs, suggesting increased T cell infiltration during glioma progression”.

We deleted “suggesting increased Tregs in GBM compared to LGG” from previous line 178.

The statement on line 215 “Overall, the percentages of macrophages were significantly higher and microglia lower in GBMs compared to LGGs (**Supp Fig 4h**)” remains factually true after reanalysis, but the statistics for h-mic was no longer statistically significant. Hence, we deleted “with the three most abundant cell types, S-Mac1, h-Mic, and i-Mic, showing statistically significant differences (**Fig 4d, Supp Fig 4c**).” from line 215.

3) Minor point, the Suva group has now published a new manuscript where they analyse also the immune cells, Hara et al Cancer Cell 2021, it would seem appropriate to cite this work and comment on how their conclusions relate the those of this manuscript.

We now reference and comment on this paper on page 19, lines 420-422.

Reviewers' Comments:

Reviewer #5:

Remarks to the Author:

The authors have adequately addressed many of my comments, however there are still two points which require attention

1) LGG-01: according to the table this is IDH wt, which the authors confirmed has been validated with sequencing and therefore it is reliable. However, diffuse astrocytomas are IDH mutant, so this case is most likely an IDH wt GBM i.e. those cases where the surgical sample does not contain the histological hallmarks of GBM but can nowadays be identified as such molecularly. The pathology report, including results of molecular tests should be reviewed.

I am afraid this an essential point which requires attention as misclassification of the cases impacts on results and conclusions.

2) Previous LGG-02 which is now ndGBM-10, please update the diagnosis in the table which still reads HG astrocytoma but it should read GBM

Rebuttal letter for NCOMMS-20-44885B

We thank reviewer #5 for thoughtful comments and below are our responses.

1) LGG-01: according to the table this is IDH wt, which the authors confirmed has been validated with sequencing and therefore it is reliable. However, diffuse astrocytomas are IDH mutant, so this case is most likely an IDH wt GBM i.e. those cases where the surgical sample does not contain the histological hallmarks of GBM but can nowadays be identified as such molecularly. The pathology report, including results of molecular tests should be reviewed. I am afraid this an essential point which requires attention as misclassification of the cases impacts on results and conclusions.

In response to reviewer comments, we reviewed the pathology report again and there was a note that it is “Grade II plus”. Considering this “soft” histological assessment and more recent molecular classification criteria the reviewer points to, we have reclassified this sample as ndGBM-11 instead of LGG-01. We reanalyzed our data with this new classification and all conclusions hold true, except that low to high grade glioma comparisons lost statistical significance due to the reduced number of LGG samples. Therefore, Fig 4d and statements referring to LGG to GBM are now deleted from the revised manuscript. Please see table below for all changes.

2) Previous LGG-02 which is now ndGBM-10, please update the diagnosis in the table which still reads HG astrocytoma but it should read GBM.

We apologize for this oversight, and it is now corrected in Supplementary Table 1.

Changes in Text:

- Added a co-author previously omitted.
- Number of samples per tumor type or sample information (LGG n=2 instead of 3 and ndGBM n=11 instead of 10 due to sample LGG-01 changing from LGG to grade IV GBM (ndGBM-11) per reviewer #5’s suggestion was changed in lines: 95
- Analysis was reperformed and statistics recalculated due to LGG-01 reassignment to ndGBM11 (samples were regrouped accordingly) in lines 130,131, 141, 170 and179
- Minor language edits for clarity, marked with track changes
- Update figure call outs after deleting Figure 4d
-

Changes in Figure Legends

- Number of samples per tumor type were changed to reflect LGG-01 to ndGBM-11 reassignment and deleted legend for Fig4d and Supp Fig 4i.

Changes in Figure		
Figure #	Change type	Panels
Fig1	Sample name change from LGG-01 to ndGBM-11	1b,c,e
	Reanalyzed the data to change in tumor type and grade: LGG-01 changed to ndGBM-11 and now grouped with ndGBMs instead of LGGs.	1f

Fig 2	Sample name change from LGG-01 to ndGBM-11	2a,b
Fig 3	Sample name change from LGG-01 to ndGBM-11	3a,e,f
	Reanalyzed data to change in tumor type and grade: LGG-01 changed to ndGBM-11 and now grouped with ndGBMs instead of LGGs.	3d
Fig 4	Sample name change from LGG-01 to ndGBM-11	4c
	Deleted LGG to GBM comparison	Previous fig4d
Fig 5	Sample name change from LGG-01 to ndGBM-11	5a,b,c,d
Supp Fig 1	Relabeled sample names to be consistent with the rest of the manuscript	sf1b,c,d,e
Supp Fig2	Sample name change from LGG-01 to ndGBM-11	sf2a,c,d
Supp Fig3	Sample name change from LGG-01 to ndGBM-11 and reorganized. Sample names updated to be consistent with the rest of the manuscript	sf3a,c,e
Supp Fig4	Sample name change from LGG-01 to ndGBM-11	sf4a,b,c,d
	Deleted LGG to GBM comparison	sf4i
Supp Fig5	Sample name change from LGG-01 to ndGBM-11	sf5a
Supp Fig7	Sample name change from LGG-01 to ndGBM-11	Sf7a,b,c,d

Reviewers' Comments:

Reviewer #5:

Remarks to the Author:

The authors have adequately addressed my comments.

Response to reviewer comments

Reviewer #5.

The authors have adequately addressed my comments.

Response: Thank you.